# The core microbiomes and associated metabolic potential of water kefir as revealed by pan multi-omics
Samuel Breselge [1,2], Iwona Skibinska [1], Xiaofei Yin[3,4], Lorraine Brennan[3,4,5], Kieran Kilcawley [1,6] & Paul D. Cotter [1,2,5] ✉

Water kefir (WK) is an artisanal fermented beverage made from sugary water, optional fruits and WK grains. WK grains can be reused to start new fermentations. Here we investigate the microbial composition and function of 69 WK grains and their ferments by shotgun metagenomics. A subset of samples was subjected to metabolomic, including volatilomic, analysis. The impact of different fermentation practices on microbial composition and fermentation characteristics was analysed and it was noted that, for example, the common practice of drying water kefir grains significantly reduces microbial diversity and negatively impacts subsequent grain growth. Metagenomic analysis allowed the detection of 96 species within WK, the definition of core genera and the detection of different community states after 48 h of fermentation. A total of 485 bacterial metagenome assembled genomes were obtained and 18 putatively novel species were predicted. Metabolite and volatile analysis show associations between key species with flavour compounds. We show the complex microbial composition of WK and links between fermentation practices, microbes and the fermented product. The results can be used as a foundation for the selection of species for large scale WK production with desired flavour profiles and to guide the regulatory framework for commercial WK production.

Water kefir (WK) is a fermented beverage, traditionally produced using WK grains, which serve as the inoculum for the associated fermenting microorganisms[1,2]. Research of, and consumer interest in, WK has been growing, including due to perceived health benefits. Despite numerous studies exploring the health benefits of WK in vitro and in animal models, there is a paucity in human interventions, as highlighted in the literature[3–9] (reviewed in ref. 10). To produce artisanal WK, WK grains are added to sugary water, to which a variety of dried or fresh fruits can be added as additional nutrient sources for the fermentation, and left to ferment at room temperature. Fermentations can be aerobic or anaerobic and last between one and several days. WK grains contain a mix of bacteria and yeast, which ferment the solution and create what can be described as a refreshing, slightly sour, slightly alcoholic, fermented beverage[11–15]. The WK grains host most of the microbes on the outer layer of the grains, which are made of the exopolysaccharide (EPS) dextran[16–18]. *Lentilactobacillus hilgardii* has been shown to be one of the key species responsible for EPS production and grain growth, but other lactic acid

bacteria (LAB) have been shown to be involved in dextran production as well[19–21]. To date, several studies have investigated either individual WKs[15,22–26], or a small number of them[12,14,27–29], to explore their microbial composition. These studies have identified LAB, acetic acid bacteria (AAB), and yeasts as the most common taxa in WK, with *Bifidobacterium* spp. and *Zymomonas mobilis* being among the most abundant microorganisms detected[12,14,15,22–29]. Despite these insights into WK microbiomes, previous studies have not had sufficient numbers of different WK to allow an investigation of the core microbiome of WK more generally or establish the existence of microbial community states across different WKs under standardized experimental conditions.

In addition to analysis of the WK microbiome, the corresponding metabolome has also been investigated. The main metabolites present, such as ethanol, different sugars and organic acids, as well as aroma compounds (volatile organic compounds), have been analysed in studies involving individual WKs and, in a few instances, a small number of WKs[23,25,26,30]. Initial steps have been taken towards recreating a WK community by

¹Teagasc Food Research Centre, Moorepark, Cork, Ireland. ²APC Microbiome Ireland, Cork, Ireland. ³UCD Institute of Food and Health, UCD School of Agriculture and Food Science, University College Dublin, Dublin, Ireland. ⁴UCD Conway Institute of Biomolecular and Biomedical Research, University College Dublin, Dublin, Ireland. ⁵VistaMilk, Cork, Ireland. ⁶School of Food and Nutritional Sciences, University College Cork, Cork, Ireland. ✉e-mail: Paul.Cotter@teagasc.ie

combining a LAB with a yeast sourced from an artisanal WK[31]. These simple microbial pairs have been able to ferment the medium, but the fermentation progressed slower and had higher residual sugar concentrations compared to fermentations with WK grains[31]. More complex, four strain communities, isolated from WK, have gained commercial interest (patent WO2023072715A1; 2023).

This study provides unprecedented insights into the microbial composition and community states of WK and identified several novel species, of which two have been already isolated[32]. The interplay between microbial composition, fermentation practices and fermentation characteristics has been investigated, providing understandings for artisanal and commercial WK producers. Microbial composition and metabolic interplay during the WK fermentation could guide the development of pitched communities for the large-scale production of WK with a rich flavour profile. We hope that these findings will guide the production of novel foods and regulatory frameworks for commercial WK.

## Results

### Sourcing water kefir grains & self-reported data
A total of 69 WK grains were sourced from 21 different countries (Fig. 1a). Of these 69 WK grains, 43 grains were shipped fresh in a small amount of liquid and 26 grains were dried at room temperature before shipping. The providers of these grains shared a variety of self-reported data relating to their fermentation practices (Supplementary Data 1). Of the 69 WK grains, eight were grains that were used commercially, 49 were from individuals that employed them for private use and information was not available (NA) for 12 grains. Many fermentations performed by these contributors were carried out for more than 24 h and up to 48 h (26/69) while others typically fermented for more than 48 h and up to 72 h (18/69) or longer than 72 h (6/69; 14 NA, Figure1b). There was a relatively even split between aerobic (30; filter or cloth allowing gas exchange) and anaerobic (27; tight lid with limited gas exchange) fermentations (12 NA, Fig. 1c). Grains had previously been used to ferment a variety of combinations of fruits (Fig. 1d). The fruit category "other" contained for example: dried pears, ginger, dates, raisins, apricots, prunes, dried apples, hibiscus tea, and orange peel. Substrates contained mostly brown sugar (Fig. 1e).

### General fermentation characteristics
The WK grains received were passed through a minimum of two initial or 'pre-' fermentations, with the intention to allow microbial communities to re-establish after shipping, before samples were taken for analysis. The starting pH of the WK media was 6.6 and the WK media, to which no grains were added, slightly increased during the 48 h of incubation. In contrast, the pH decreased to an average pH of 5.3 after 8 h of incubation with inoculum and further decreased to an average of 3.6 after 48 h of fermentation (Fig. 1f).

The kefir grains that exhibited the strongest grain growth increased in mass by up to 236% during 48 h of fermentation. Grains that were received in fresh form showed a significantly stronger grain growth than grains that were dried before shipment. A strong increase (79.0% mean; 77.9% median) in grain mass was observed in grains that were shipped fresh compared to grains that were send dried (18.2% mean; 0.44% median) after 48 h of fermentation (Fig. 1g and Supplementary Fig 1). Grain growth was not significantly impacted by freezing (see Supplementary Results and Discussion; Supplementary Fig 2).

### Metagenome assembled genomes (MAGs) and detection of putatively novel species
For each of the 69 WK grains, five samples were the subject of shotgun sequencing (2 × 8 h liquid, 2 × 48 h liquid, 1 × 48 h grains). The raw sequencing data was shared within the MASTER consortium, to be integrated into the curatedFoodMetagenomicData (cFMD) database[33]. Here, the sequencing data has been analysed with a WK specific MAG assembly (co-assembly) and custom taxonomic profiling approach.

Additional sequencing information can be found in Supplementary Results and Discussion and in Supplementary Data 1.

MAG assembly yielded 485 high quality bacterial MAGs. 54 of these MAGs could not be assigned to the species level using a 95% average nucleotide identity (ANI) cut-off (Fig. 2a, Supplementary Data 1), suggesting they are potentially novel species. Some of these MAGs share more than 95% ANI with each other, suggesting that these are the same novel species, detected in WKs from different sources. After de-replication of the unassigned MAGs at 95% ANI, a total of 18 MAGs representing 18 putatively novel species remained. The putatively novel species belong to the genera *Acetobacter* (4x), *Bacillus* (1x), *Bifidobacterium* (2x), *Clostridium* (1x), *Ethanoligenens* (1x), *Gluconobacter* (1x), *Liquorilactobacillus* (1x), *Oenococcus* (1x), *Propionibacterium* (1x), *Pseudoclavibacter* (1x), and *Sporolactobacillus* (4x). Hereafter, the putatively novel species will be referred to by their genus name, followed by a WK number in which it was found and MAG number (e.g., *Liquorilactobacillus WK059_bin.2*). One of the novel *Bifidobacterium* species was detected in 13 different WKs, while the other species was only detected in a single WK. Isolates corresponding to these novel *Bifidobacterium* species were successfully cultured and characterized[32].

A total of 97 fungal MAGs with ≥70% BUSCO completeness were obtained. The MAGs were assigned to the species *Brettanomyces anomalus* (n = 6), *Br. bruxellensis* (n = 4), *Lachancea fermentati* (n = 8), *Pichia fermentans* (n = 3), *Pi. membranifaciens* (n = 4), *Saccharomyces cerevisiae* (n = 51), *Sa. uvarum* (n = 1), *Torulaspora quercuum* (n = 2), *Zygotorulaspora florentina* (n = 18; Fig. 2b).

### Taxonomic profiling
A custom WK specific inStrain database was used to identify the species present across the WK pangenome. To consider a species a true positive hit, a strict cut off with a breadth of at least 0.35, as well a minimum ratio of 0.75 for the expected breadth to observed breadth, was required. No species were detected in the control media samples with these requirements. Ultimately, an average of 10.3 species were detected per WK sample (Supplementary Fig 3). A total of 96 species, from 28 genera, were detected using inStrain and 76 of these species were supported with at least one MAG (Fig. 3a, b; Supplementary Table 1).

The six most commonly detected species, prevalent in more than 50% of the 69 WKs, were *Saccharomyces cerevisiae* (85.5% prevalence), *Lacticaseibacillus paracasei* (75.4%), *Liquorilactobacillus satsumensis* (71.0%), *Lentilactobacillus hilgardii* (65.2%), *Acetobacter orientalis* (53.6%), and *Liquorilactobacillus nagelii* (50.7%) (Fig. 3a). The most prevalent genera were *Saccharomyces* (85.5%), *Gluconobacter* (84.1%), *Liquorilactobacillus* (84.1%), *Acetobacter* (82.6%), *Lacticaseibacillus* (75.4%), *Lentilactobacillus* (69.6%), and *Bifidobacterium* (55.1%), being prevalent in more than 50% of the 69 WKs tested. *Leuconostoc* (43.5% prevalence), *Zygotorulaspora* (40.6%), *Pichia* (39.1%), *Brettanomyces* (36.2%), *Oenococcus* (34.8%), and *Zymomonas* (33.3%) were also commonly detected in WK (Fig. 3b).

While *Sa. cerevisiae* was the most prevalent species, it was often not the most abundant species (Supplementary Fig 4). Indeed, although *Sa. cerevisiae* has an average abundance of 9.05% across the samples in which it was detected, *Zymomonas mobilis* has an average abundance of 35.62% in the samples in which it was detected. The highest abundance of a single species in a sample was *Li. satsumensis* with 97.98% (sample WK065-2-G-48h). Some samples showed a high abundance of *Bifidobacterium*, especially at the 48 h time point, with up to 63.55% relative abundance (*Bi. aquikefiri* max. abundance 55.56% in WK004-2-G-48h; *Bi. psychraerophilum* 49.00% in WK007-2-G-48h; *Bi. tibiigranuli* 25.51% in WK012-2-L-48h; *Bi.* WK041_bin.7 63.55% in WK041-2-L-48h; *Bi.* WK044_bin.6 31.75% in WK044-2-G-48h; Supplementary Table 1).

### Alpha diversity analysis
A significant increase in alpha diversity (Shannon index, but not Simpson index), within sample diversity, was noted in the liquid samples after 8 h and 48 h of fermentation compared to grain samples (Fig. 4a, b). A decrease in

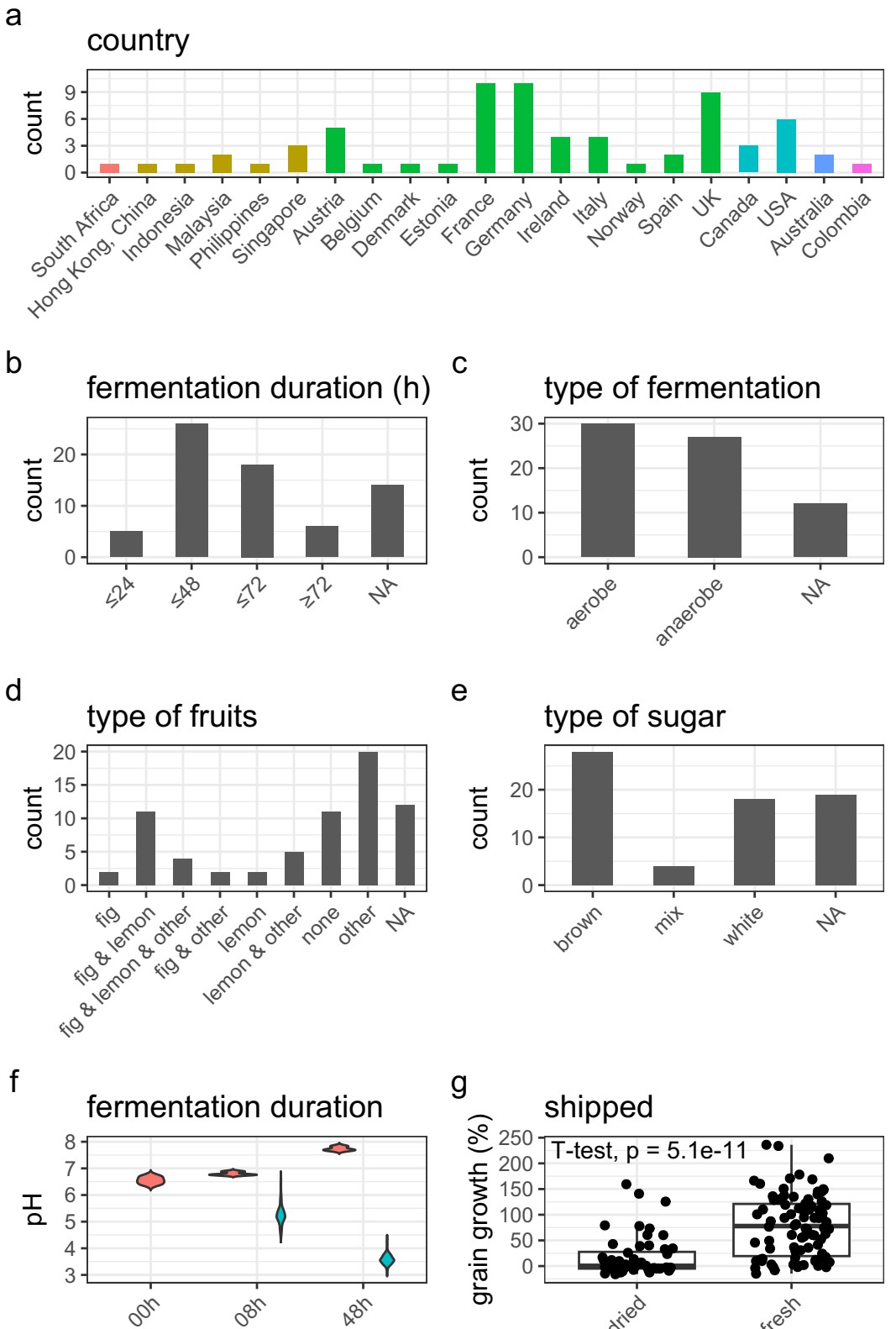

**Fig. 1 | Water kefirs from around the world, have previously been fermented under diverse conditions, show similar changes in pH, but their grain growth is influenced by the way grains were prepared for shipment. a** Counts of WK grains that were sent to us and their respective 21 countries (colour coded by continent). Self-reported fermentation practices provide insights into most common practices. **b** Most commonly, WKs are fermented for more than 24 h and up to 48 h, while 48 h to 72 h fermentations are common as well. **c** Providers report a near even split of aerobic and anaerobic fermentations. **d** Fermentations are done with a variety of different fresh and dried fruits. **e** Brown sugar is the most common sugar used. **f** The pH of the media or the fermentation was measured at the start of the fermentation (media only), 8 h, and 48 h of fermentation. Media $n = 3$ (experimental replicates, red violins); WK $n = 138$ (69 WK with two experimental replicates each; blue violins). **g** WK grains that had been dried for shipment show a decreased grain growth, compared to grains that were sent fresh. Dried $n = 52$ (26 WK with two experimental replicates each); fresh $n = 86$ (43 WK with two experimental replicates each). NA not available.

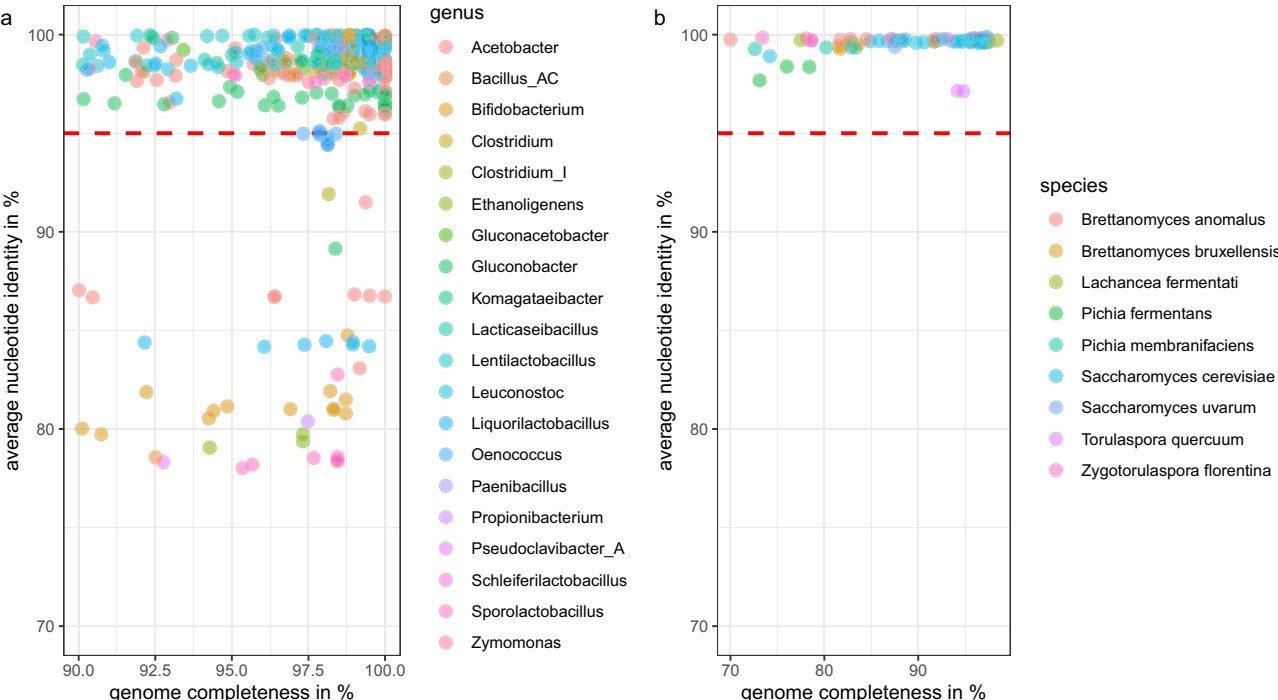

**Fig. 2 | Bacterial and fungal metagenome assembled genomes (MAGs) identifies several putatively novel species. a** Bacterial high quality MAGs (*n* = 485) were assigned taxonomy at genus level using GTDB-Tk[71] and genome completeness was assessed using CheckM[69]. The average nucleotide identity (ANI) was calculated using FastANI[72] to the RefSeq or GenBank genome with the closest

ANI. **b** Fungal MAG completeness was assessed using BUSCO[81]. A total of 97 fungal MAGs showed a genome completeness of ≥70%. The average nucleotide identity (ANI) was calculated using FastANI[72] to the RefSeq and GenBank genome with the closest ANI. FastANI was used for taxonomic species assignment.

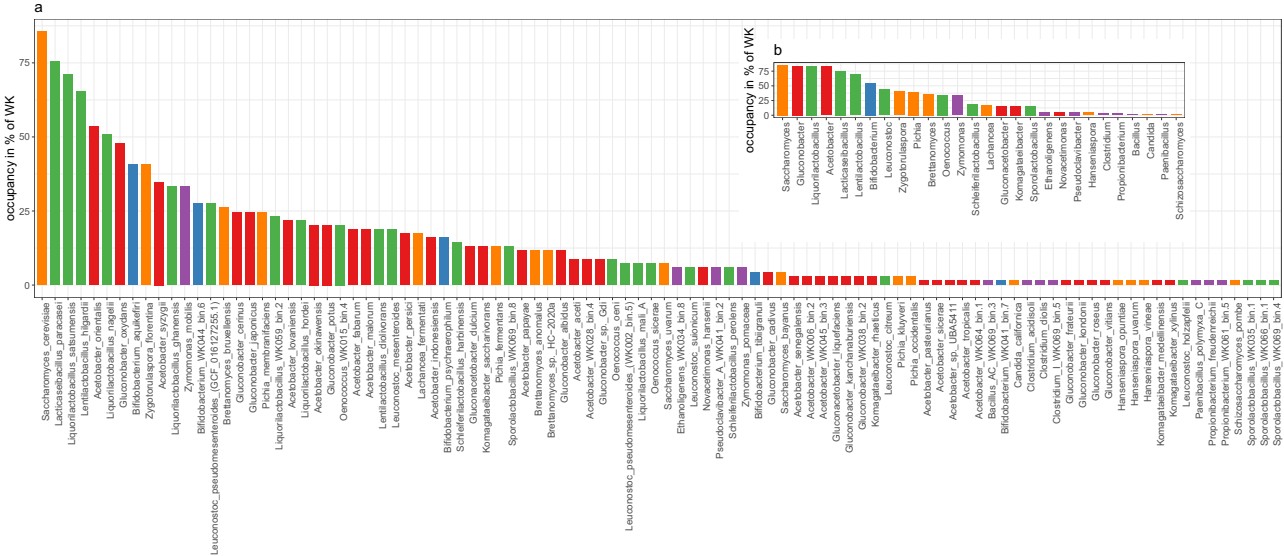

**Fig. 3 | Microbial composition of water kefir. a** Species or **b** genera were counted as present if they were detected with inStrain[73] in at least one of the five samples of each of the 69 WK. Yeasts are indicated in orange, LAB in green, AAB in red, bifidobacteria in blue, and other taxa in purple.

alpha diversity was noticed for samples derived from grains that were dried for shipment, compared to samples from grains that were sent fresh (Fig. 4c, d). Several significant effects from the previous fermentation practices persisted under the controlled fermentation conditions in the laboratory (Supplementary Fig 5). E.g. a deceased alpha diversity was observed in samples with previous aerobic fermentations compared to anaerobic fermentations (Supplementary Fig 5a, b) and the highest alpha diversity was observed in samples that were previously fermented for ≥72 h (Supplementary Fig 5c, d).

## Water kefir community states

After 48 h of fermentation, hierarchical clustering revealed six WK community states (CS), with CS clustering primarily by the most abundant genera (Fig. 5a). States were dominated by *Liquorilactobacillus* (CS 1), *Gluconobacter* (CS 2), *Zymomonas* (CS 4), *Leuconostoc* (CS 5), and *Acetobacter* (CS 6), or were relatively balanced communities without a clearly dominant genus (CS 3, Fig. 5a). Different CS were supported by genus-level beta diversity analysis (Fig. 5b; ANOSIM statistic R: 0.6967), while species-level beta diversity analysis shows a less clear separation between the

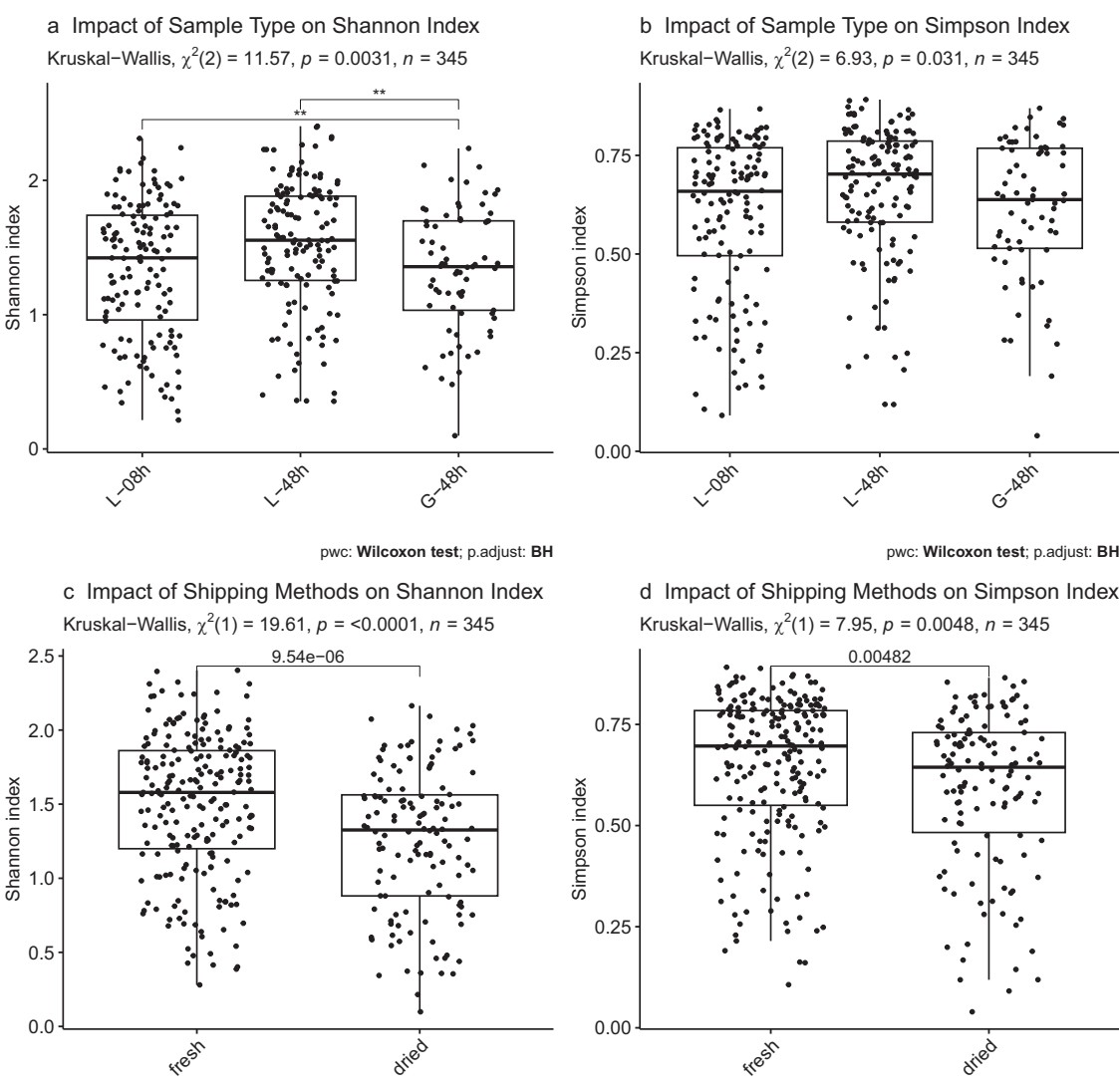

**Fig. 4 | Impact of sample type and shipping method on alpha diversity measures.** Alpha-diversity was increased in the 48 h liquid samples, while drying of WK grains reduced alpha-diversity measures. **a**, **b**, Alpha diversity of grain samples taken after 48 h of fermentation (G-48h, $n = 69$); liquid samples taken after 8 h of fermentation (L-08h, $n = 138$); liquid samples taken after 48 h of fermentation (L-48h, n = 138). **c**, **d**, Alpha-diversity of samples that were sent to us either dried or fresh. Dried $n = 130$ (26 WK x 5 samples); fresh $n = 215$ (43 WK x 5 samples).

proposed community states (Fig. 5c; ANOSIM statistic R: 0.3807). *Zymo-monas* dominated WKs show a separate CS on genus- and species-level beta diversity analysis (Fig. 5b, c).

**Correlation analysis of species with alpha diversity measures and fermentation characteristics**
Significant correlations were observed between various species and alpha diversity measures, as well as fermentation characteristics, such as pH and relative grain growth (Fig. 6). *Len. hilgardii, Liquorilactobacillus WK059_bin.2, Oenococcus WK015_bin.4, Zyg. florentina, Lact. paracasei, Bi. aquikefiri*, and *Li. nagelii* correlated positively with the different alpha diversity measures and relative grain growth. Several *Acetobacter* spp. show significant negative correlations with pH (high abundance at low pH) and positive correlations with alpha diversity measures. *Sa. cerevisiae* shows strong positive correlations with alpha diversity measures, while *Zym. mobilis* shows some of the strongest negative correlations with alpha diversity measures, as it usually highly abundant and dominating when it is present in a WK community.

**Metabolite analysis**
A selection of 15 WK fermentations, representing a range of microbial compositions, were further investigated by NMR metabolomic and GC-MS volatile organic compound (VOC) analysis at 8 h and 48 h. Metabolomics detected 29 compounds in the WK liquid (Supplementary Data 1), with the majority of samples clearly separating on the basis of whether they were collected at 8 h or 48 h (Supplementary Fig 6). There was an overall pattern of decreased sugars during the sampled time points of the fermentation, while alcohols and organic acids increased during the fermentation (Supplementary Fig 7 and Supplementary Fig 8). The WK media contained 80 g/L sucrose and after autoclaving 61.9 g/L sucrose (180.7 mM), 6.1 g/L glucose (34.0 mM), and 5.7 g/L fructose (31.4 mM) were detected by NMR analysis. Through the fermentation, sucrose concentrations were reduced to 42.5 g/L on average, meaning that 69% of the sucrose remained after the 48 h of fermentation. On average 0.647 g/L lactate (7.3 mM), 0.002 g/L acetate (0.03 mM) and 0.964% ABV or 7.61 g/L ethanol (165.12 mM) were produced during the 48 h of fermentation (Fig. 7). Out of the 29 metabolites detected, 22 differed significantly during the sampled time points of the fermentation (Supplementary Fig 7). Acetoin, citrate, ethanol, fructose,

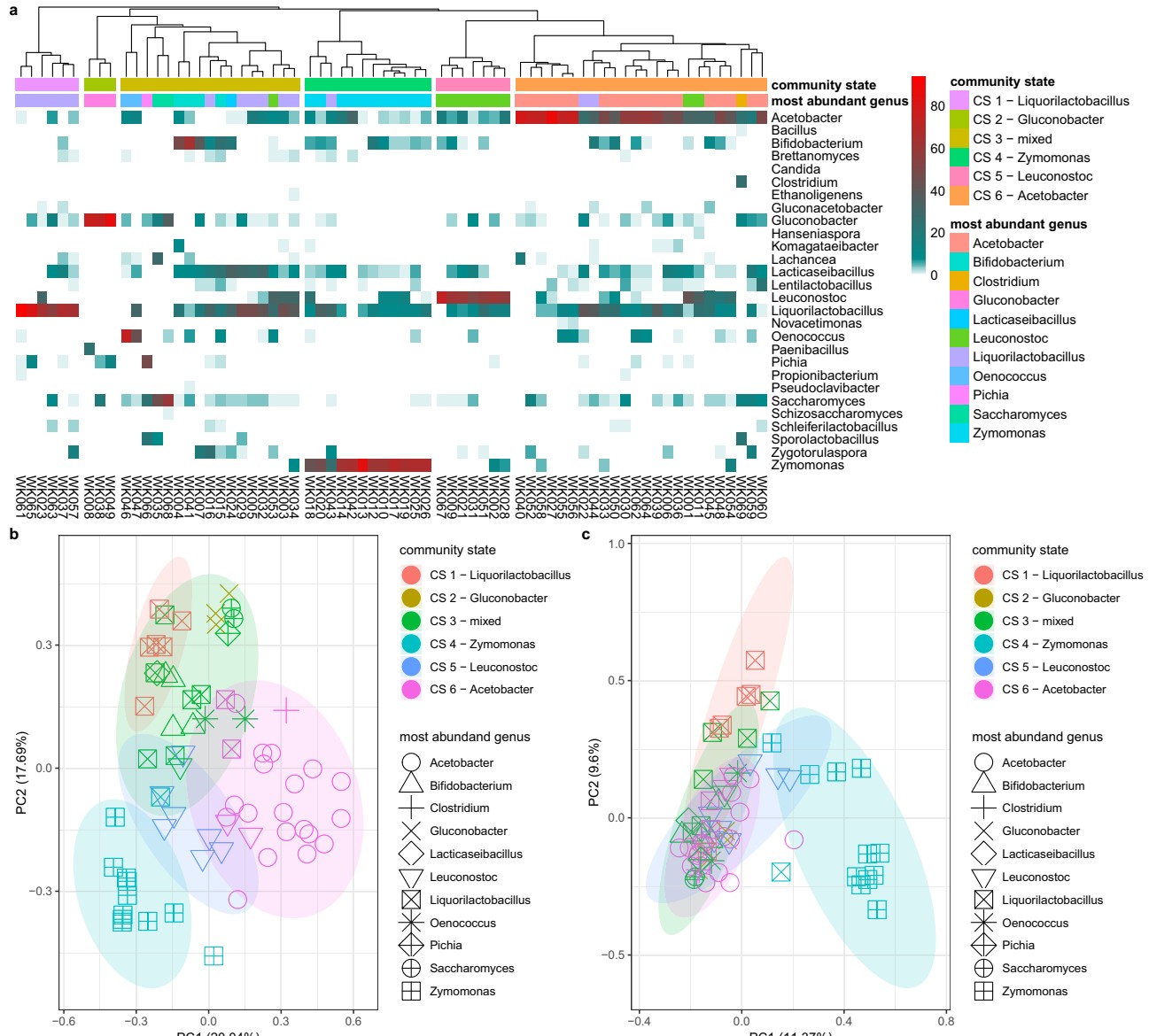

**Fig. 5 | Water kefir show six community states after 48 h of fermentation.**
**a** Average relative abundance of genera after 48 h of fermentations suggests different community states (CS) in WK. **b** Genus-level beta diversity analysis of 48 h fermentation samples shows separation by CS (ANOSIM statistic R: 0.6967, Significance: 1e−04). Beta diversity using Bray-Curtis distance, based on average

relative abundance of genera after 48 h of fermentations (*n* = 69). **c** Species-level beta diversity analysis of 48 h fermentation samples shows less clear separation by CS (ANOSIM statistic R: 0.3807, Significance: 1e−04). Beta diversity using Bray-Curtis distance, based on average relative abundance of species after 48 h of fermentations (*n* = 69).

isoleucine, lactate, methylamine, propylene glycol, pyruvate, pyruvatoxime, and valerate accumulated during the fermentation while choline, formate, fumarate, 4-aminobutyrate (GABA), malate, proline, succinate, and sucrose decreased (Supplementary Fig 7). Acetate was elevated at 8 h, but then decreased again after 48 h of fermentation (Supplementary Fig 7). Alanine and leucine concentrations decreased during the first 8 h of the fermentation and then increases again towards 48 h of fermentation, to not significant levels compared the media controls (Supplementary Fig 7). No significant differences between the media control, the 8 h and 48 h fermentations was detected for glucose, glutamine, glycylproline, homocitrulline, malonate, methanol, and valine. It should be noted though, that some WKs showed the production or degradation of some of these compounds while not being overall statistically different. *A. orientalis* showed significant positive correlations with isoleucine and leucine (Fig. 8). *Gluconobacter oxydans* showed positive correlations with pyruvate and *Lach. fermentati* showed positive correlations with homocitrulline. *Li. satsumensis* and *Lact.*

*paracasei* showed nearly significant correlations with lactate (adj. p-value 0.064 and 0.094, respectively). None of the negative correlations were statistically significant after p-value adjustment.

## Flavour analysis

Volatile organic compounds (VOCs) are contributors to the flavour and their analysis, a.k.a. flavour analysis, allowed the detection of 84 compounds (Supplementary Data 1). Media samples clustered separately from fermented samples and a grouping of the WK samples by time point was discernible (Fig. 9 and Supplementary Fig 9). At 8 h of fermentation volatile alcohols increased most, while volatile acids and esters increased by 48 h of fermentation (Supplementary Fig 10). After 48 h of fermentation, the *Zym. mobilis* dominated WK (WK042) showed a distinct VOC profile, rich in esters, and clustered separately from other WK flavour profiles (Fig. 9). Out of the 84 detected VOCs, 48 VOCs showed a significant difference during the fermentations, while 25 VOCs did not significantly change during the

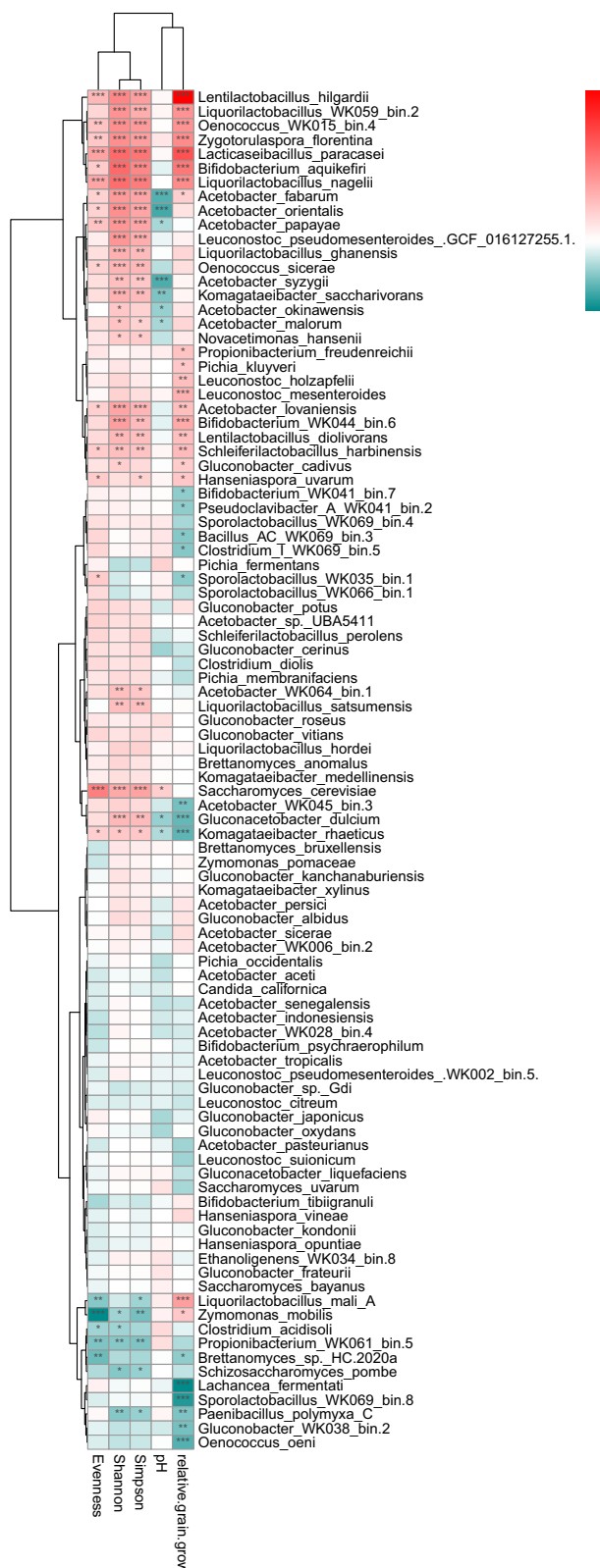

**Fig. 6 | Water kefir microbes and their correlation with fermentation characteristics.** Correlation analysis indicates species correlating with alpha diversity measures and fermentation characteristics. Spearman correlation analysis of species abundance with alpha diversity measures (Evenness, Shannon index and Simpson index), pH, and relative grain growth by weight.

sampled time points of the fermentation. Eleven compounds were not statistically tested, as two of the conditions did not contain detectable quantities of the VOC (Supplementary Fig 11). Eight of the eleven not statistically tested compounds were present only in the 48 h fermentations and three were only detected in the 8 h fermentations.

Correlation analysis showed 40 significant positive and 9 significant negative correlations between VOCs and species that were detected through shotgun metagenomics. *Br. bruxellensis* and *Brettanomyces sp.* HC-2020a formed a cluster with strong positive correlations with several esters, such as ethyl 2-methylbutanoate, ethyl tetradecanoate, isobutyl octanoate (*Br. sp.* HC-2020a only), methyl dodecanoate (*Br. bruxellensis* only), methyl decanoate (*Br. bruxellensis* only), and ethyl dodecanoate (*Br. bruxellensis* only; Fig. 10). Many of these esters are described as fruity and sweet. *Pi. fermentans* is the only species that showed significant negative correlations with this cluster of esters. Overall, *Pi. fermentans* contributes to the development of the VOC profile of WK in a particular way, as it showed eight significant negative and six significant positive correlations with VOC flavour compounds. *Pi. fermentans* had positive correlations with acetone, 2-methylfuran, ethyl ether, 2-butanone, 2-methylpropanal, and hexane (Fig. 10). *Sa. cerevisiae* showed positive correlations with styrene and a negative correlation with acetone. *Zyg. florentina* showed strong positive correlations with 2-nonanone. *Zym. mobilis* showed significant positive correlations with methyl dodecanoate and methyl isobutyl ketone (Fig. 10). Multi-factor analysis supports these trends (Supplementary Fig 12). Positive correlations between volatile esters and *A. indonesiensis*, *A. orientalis*, *A.* WK045_bin.3, *Bi.* WK041_bin.7, *Lach. fermentati*, *Pseudoclavibacter* A WK041_bin.2 and *Zym. mobilis* were observed as well.

## Discussion

WK is a fermented beverage produced with a diverse set of fermentation practices and microbes[2]. We obtained 69 WK grains from 21 different countries (Fig. 1a) to study WK in an unprecedented depth.

We asked the providers of the grains how they performed their WK fermentations in order to identify the most common WK fermentation practices and to guide our experimental approaches. The self-reported data from our participants suggested that most WKs are fermented for 2 to 3 days (Fig. 1b) and therefore we chose to collect our endpoint samples at 48 h, as well as another sample representative of the early stages of the fermentation (8 h). There was a relatively even split between aerobic and anaerobic fermentations among the participants (Fig. 1c). This is an interesting feature of WK as other fermented beverages typically have specific aerobic or anaerobic needs, e.g., kombucha and vinegar fermentations are performed under aerobic conditions (reviewed in refs. 34,35), while wine, beer, and cider fermentations mostly require anaerobic conditions to allow *Sa. cerevisiae* to produce ethanol and to reduce the growth of undesired microbes, such as AAB[36]. Aerobic WK fermentations generally favour AAB growth, increased acetic acid concentrations, decreased lactic acid and ethanol concentrations, and reach a lower pH[15]. Aerobic fermentations were chosen for the experimental conditions with a view to the production of beverages with low alcohol content. The reported use of different substrates during the fermentation (Fig. 1d, e) matches the observations in previously published reviews[1,2], highlighting the diversity across WK fermentation practices. How the availability of different nutrients can influence the fermentations has been investigated by Laureys et al.[15], e.g. elevated nutrient concentrations favoured the growth of *Sa. cerevisiae* and *Li. nagelii*.

The decrease of the pH during the fermentation was used as an indicator of recovery of the grains after shipping and of a successful fermentation (Fig. 1f). A drop in pH to 3.6 after 48 h was comparable to previously reported WK fermentations[14,15]. We observed that different inocula strongly impacted grain growth. At the extremes, some grains lost mass during fermentations, while others gained more than 200% (Supplementary Fig 1; Fig. 1g). Previous studies have shown that the inoculum can impact grain growth, with reported growth ranging from 4.58 to 63.82% across three

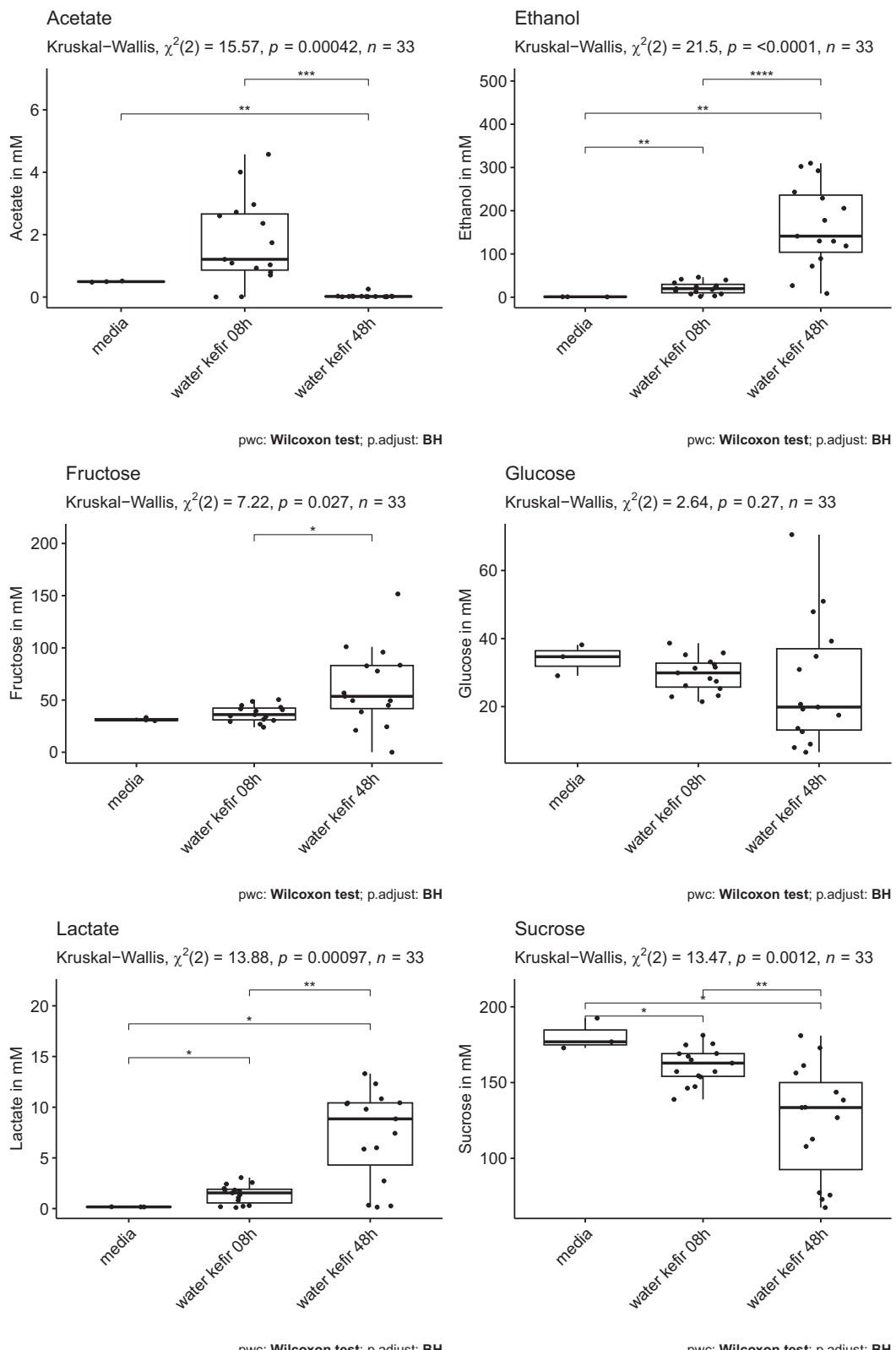

**Fig. 7 | Boxplot of selected metabolites showing statistical differences between metabolite abundances during the sampled time points of the fermentation.** Media $n = 3$; WK 08 h $n = 15$, WK 48 h $n = 15$.

different WK[14]. Associations between WK grain growth, species, and storage conditions (drying and freezing), are discussed in the Supplementary Results and Discussion.

MAGs can be used to identify putatively novel species in fermented foods[29,37]. The commonly accepted threshold of <95% ANI for bacterial species delineation[38] allowed the detection of 18 putatively novel species (Fig. 2), with their biology remaining to be explored. These previously undescribed species make up nearly 20% of the WK-associated species detected in this study, highlighting the potential for discovering novel organisms from this fermented beverage. We set out to validate our

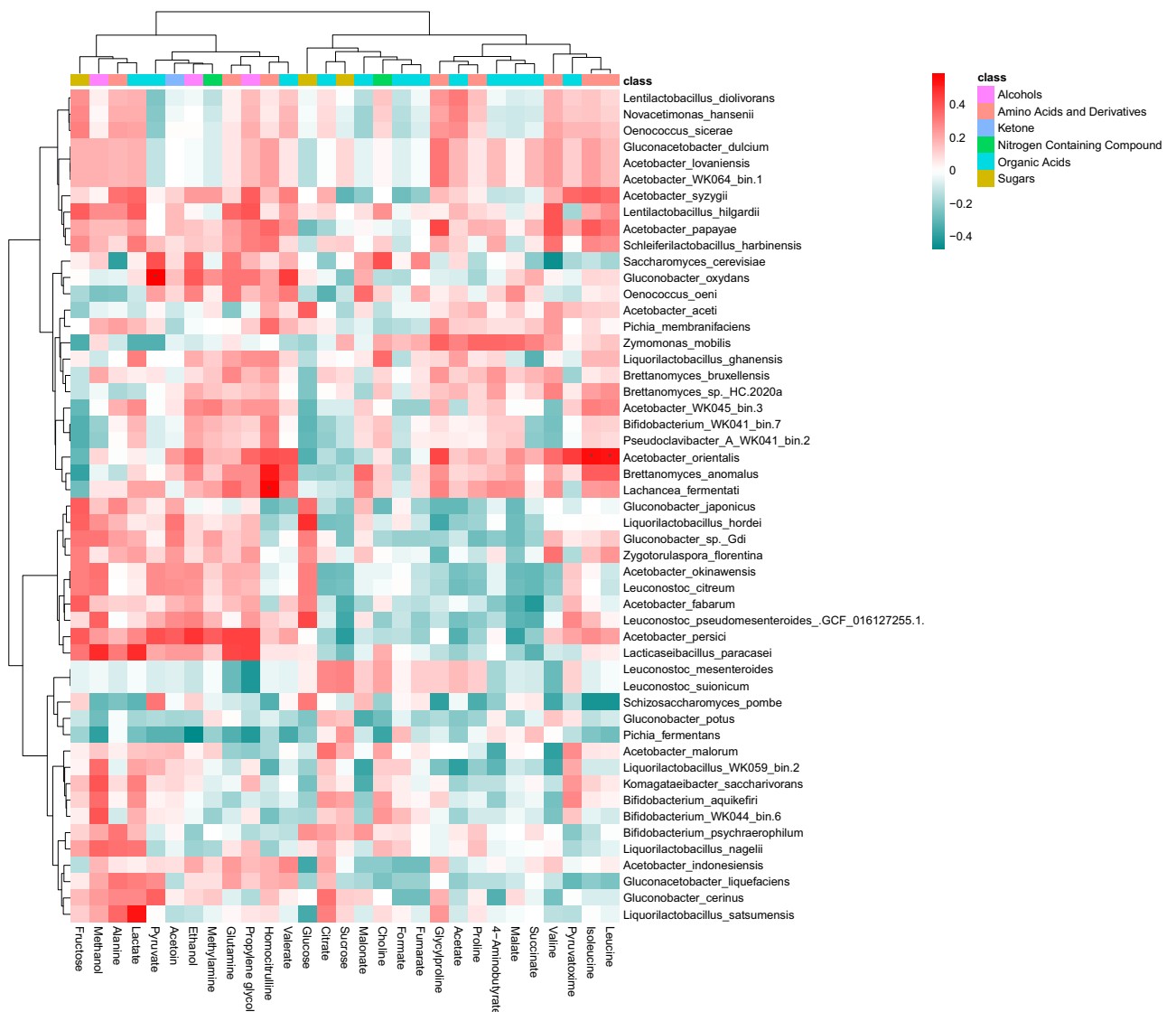

**Fig. 8 | Water kefir microbes and their correlation with metabolite production.** Spearman correlation analysis of 15 WKs with one 8 h and one 48 h sample and their species abundance and metabolite profile indicates key species influencing the WK metabolite profile.

predictions and successfully isolated the two novel *Bifidobacterium* species[32], extending the number of bifidobacteria that have been isolated from WK[39,40]. Genomic standards for fungal species delineation are the subject of controversy[41], but a similar ANI value of 95% has been suggested for possible species delineation in yeasts[42] and no putative novel yeasts were detected in this study.

To our knowledge, more than half of the species detected in this study, have not been isolated from WK before (Supplementary Table 1), highlighting a reservoir of microbes that could be explored for the production of fermented foods and other applications. The use of starter cultures for the development of novel fermented foods have been discussed by Gänzle et al.[43]. The particular role of *Zym. mobilis* and the presence of potentially undesired species, such as *Clostridium* spp. are discussed in the Supplementary Results and Discussion.

Most WK studies have analysed the microbial composition of a single WK community or a small number of communities[12,14,15,22–29]. As these studies have used different methods for species detection, such as culture based and culture independent methods with different protocols, it is difficult to compare the results and to estimate the prevalence of species within the different WK communities. This study allows unprecedented insights into prevalent taxa within WK, providing a foundation for defining a WK core microbiome. Core microbiomes are commonly defined either by

abundance or occupancy of taxa, or through combinations of these two[44,45]. A recent study by Custer et al., 2023[46] has suggested that occupancy based methods are most reliable for defining core microbiomes and using genus level taxonomic information is a common approach for this in metagenomic studies[45]. We observed that drying of WK grains for shipping, compared to shipping fresh grains, has a negative impact on the abundance of several species and genera (Supplementary Fig 13 and Supplementary Fig 14). We therefore propose a minimum 30% occupancy for defining the WK core microbiome. Following this definition, the WK core microbiome consists of yeasts (*Saccharomyces, Zygotorulaspora, Pichia, Brettanomyces*), AAB (*Gluconobacter, Acetobacter*), LAB (*Liquorilactobacillus, Lacticaseibacillus, Lentilactobacillus, Leuconostoc, Oenococcus*), as well as *Bifidobacterium* and *Zymomonas*. The here defined WK core microbiome and the average of 10.3 species per sample provides a foundation for defining what a synthetic or pitched WK community for industrial production could look like. We hope that this work will contribute towards a regulatory definition of WK, similar to the definitions that have been in place for other fermented foods, such as yoghurt and milk kefir (reviewed in ref. 47).

Different definitions and approaches have been proposed to define community structures, such as community types, community states (CS), enterotypes and community signatures[48–53]. We chose to investigate CS based on a genus-level abundance-based approach after 48 h of

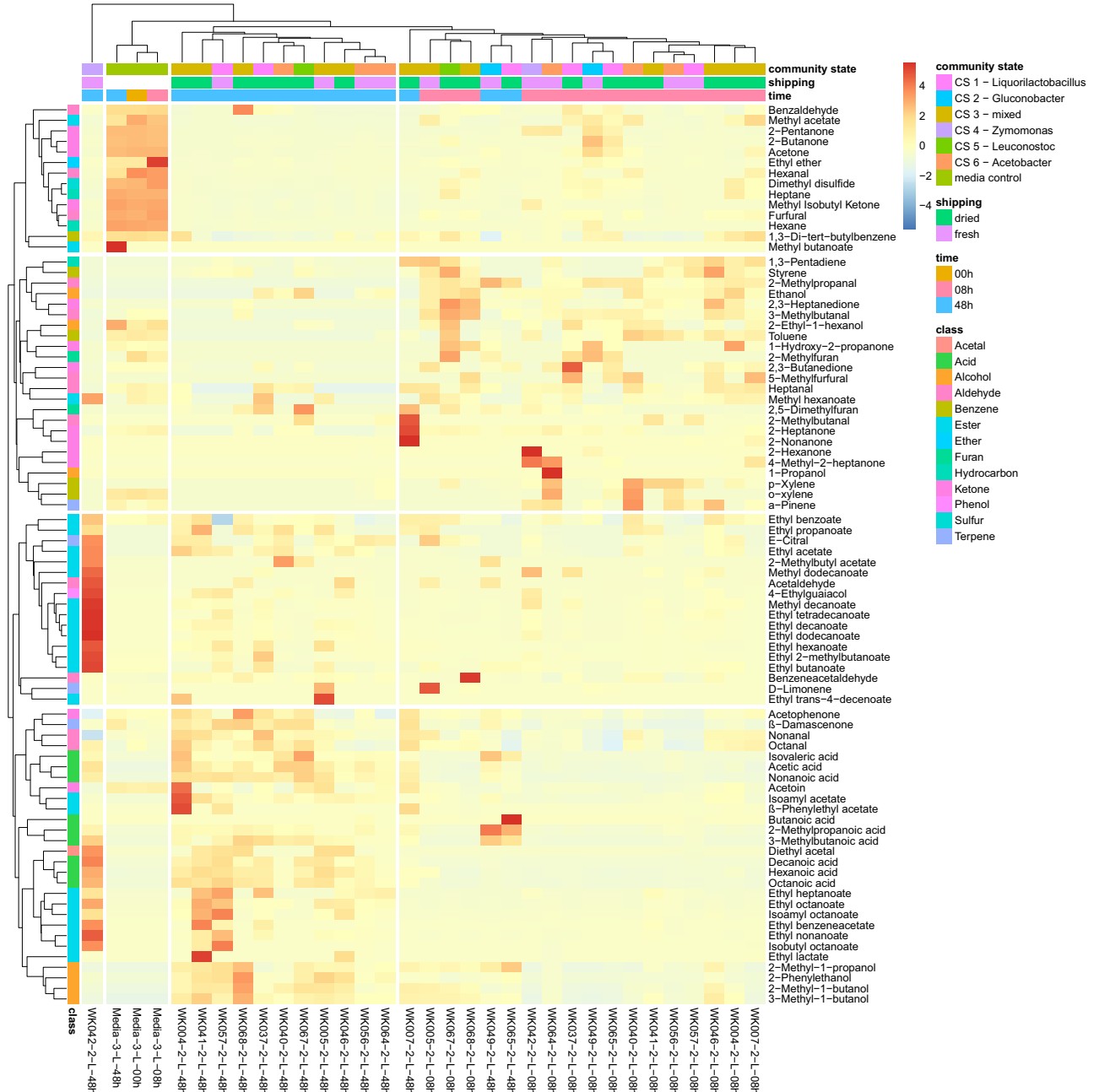

**Fig. 9 | Heat map detected volatile organic compounds.** Heat map of volatile organic compounds (VOCs) that were detected by GC-MS analysis, scaled by VOC.

fermentation, as it closest reflects the different CS present in the consumed beverages. After 48 h of fermentation, six different CS were discernible at genus level by hierarchical clustering and beta diversity analysis (Fig. 5). In contrast, milk kefir community types were driven by four different species, from only two genera[54]. It is quite plausible and logical that different WK fermentation practices allow WK communities to transition between community states. For example, LAB species are abundant at the early fermentation stages (Supplementary Table 1, ref. 13), while long and aerobic fermentations favour the growth of AABs (Supplementary Table 1[15]). Therefore, repeated short fermentations could lead to a shift in the community away from AABs towards LABs. Similar temporal trends have been observed in milk kefir as well[55] and Laureys et al. showed how WK microbial compositions changed based on nutrient availability[15]. We therefore suggest the use of the term *community states* (CS) rather than *community types* with respect to WK. In addition to the fluid nature of WK community states, it should be noted that there are likely other factors defining microbial

compositions, such as the occurrence of different microbial compositions at different geographic locations, as discussed elsewhere[1].

The community composition changed during the sampled time points of the fermentation, with an increase in alpha diversity (Shannon index), between the grain samples and the 48 h liquid samples, as well as between the 8 h liquid samples and the 48 h liquid samples (Fig. 4a, b). Patel et al.[23] observed that the WK grain community was relatively stable, while the liquid community changed rather strongly during the sampled time points of the fermentation. The drying of WK grains for shipping lead to a strong reduction in diversity (Fig. 4c, d). Additionally, we observed influences of fermentation practices (self-reported data from grain providers) on alpha diversity measures that persisted throughout the shipment of the grains and the pre-fermentations (Supplementary Fig 5), highlighting complex interactions between fermentation practices, individual species, and microbial compositions (Fig. 6 and Supplementary Fig 5).

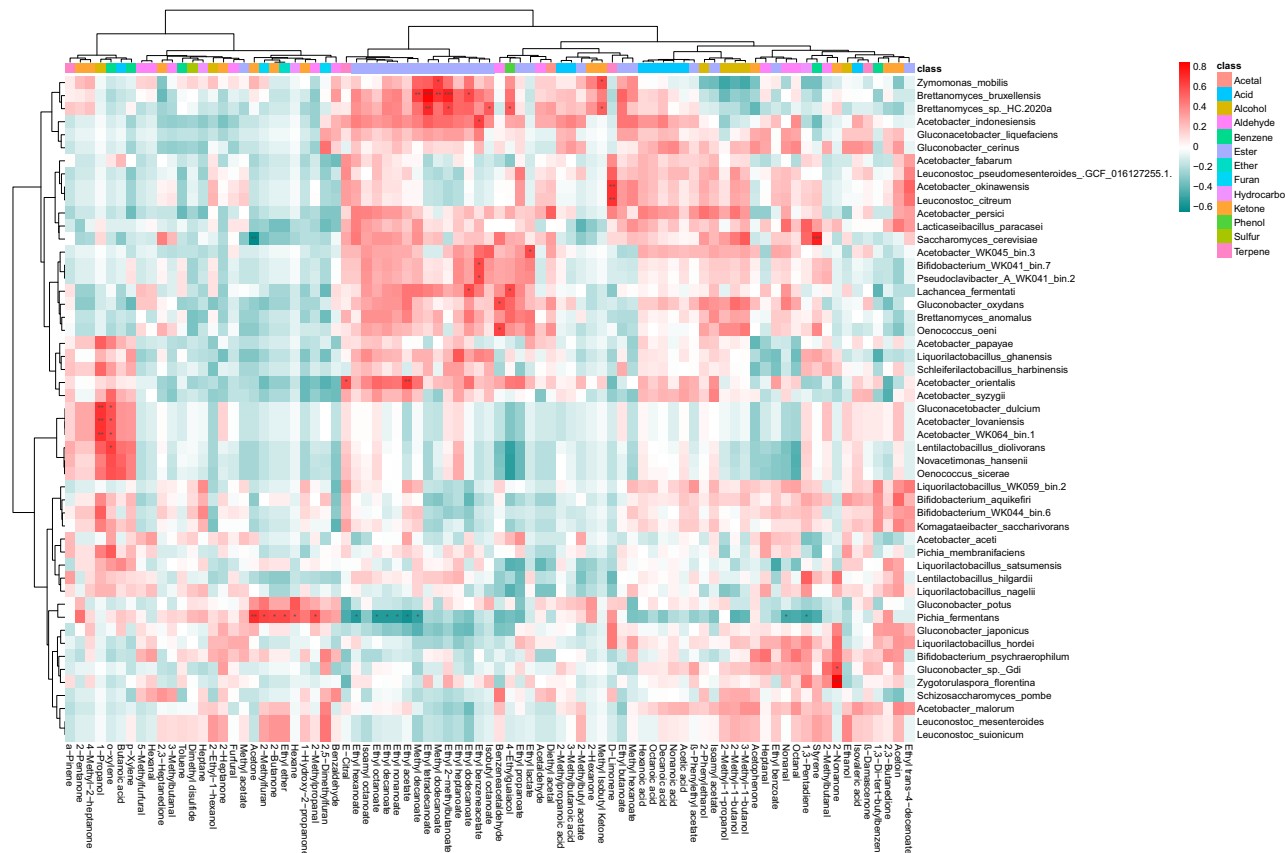

**Fig. 10 | Water kefir microbes and their correlation with flavour development.** Spearman correlation analysis of 15 WKs with one 8 h and one 48 h sample and their species abundance and VOC profile. Correlation analysis indicates key species, such as *Brettanomyces* spp., *Zymomonas mobilis*, and *Pichia fermentans*, influencing the WK volatile organic compound (VOC) profile.

The high amount of remaining sugars (Fig. 7, Supplementary Fig 8d) suggests that fermentations could have been further extended if the production of a beverage with lower sugar content was desired. How various factors can influence the fermentation progress, such as temperature, microbial composition of the inoculum, amount of inoculum, starting concentration of sucrose, and other compositional factors in the WK medium has been investigated before[14,15,25,56]. In this study, the average production of lactate and acetate after 48 h of fermentation are relatively low and the ethanol production is relatively high compared to some reports in the literature[14,15,25,56]. At the same time, these values are quite in line with other studies[23], highlighting the different fermentation practices and WK products. Organic acids are primarily produced by LAB and AAB, while ethanol is primarily produced by yeasts and *Zymomonas* (reviewed in ref. 1), suggesting that the experimental setup might have favoured the growth of yeasts and *Zymomonas*. Regulations around non-alcoholic beverages vary between countries, but for example, in the European Union and the USA 1.2 and 0.5% ABV are allowed in non-alcoholic beverages, respectively (ref. 57 and ref. 58). Some of the WK in this study exceed the legal limits for non-alcoholic beverages, suggesting the importance of alcohol-level testing in water kefir. It should be noted that a change in fermentation practices, such as additional aeration, could have further increased the growth of AABs leading to an increased production of organic acids and reduction of ethanol[15,59]. The possible detection of pyruvatoxime has been discussed in the Supplementary Results and Discussions.

*A. orientalis* showed positive correlations with the branched chain amino acids leucine and isoleucine, suggesting it is a possible producer of these amino acids (Fig. 8). Patel et al.[23], observed a similar correlation between *A. indonesiensis* and leucine in WK. Metabolic exchange is common among microbes[60]. For example, *Li. hordei* and *Li. nagelii* have previously been shown to be auxotroph for several amino acids, including leucine and isoleucine, and to benefit in a co-culture system in which yeasts provide these amino acids to the *Liquorilactobacillus* spp[61]. Our observations could suggest possible mutualism of *Liquorilactobacillus* spp. and *A. orientalis*, but while we did not observe a significant co-occurrence of *Li. hordei* or *Li. nagelii* with *A. orientalis*, we did observe a positive co-occurrence of *Li. satsumensis* and *A. orientalis* (Supplementary Fig 15). The lack of statistically significant correlations between metabolites and individual species could indicate a redundancy in metabolic capacity among individual species.

Volatile organic compound (VOC) detection can be used to assess food quality and safety[62], and further more can be used as a way to monitor the progression of food fermentations[63,64]. In WK, we saw an increase in volatile alcohols after 8 h of fermentation and the volatile profile was dominated by alcohols, acids, and esters after 48 h of fermentation (Supplementary Fig 10), leading to a clear separation of the volatile profiles (Fig. 9 and Supplementary Fig 9). An increase in VOCs accumulation during WK fermentations has been reported before[23,30]. In general, yeasts and LABs are key producers of esters (created from an acid and an alcohol), while yeasts are also able to produce complex alcohols[64], and volatile fatty acids can be produced by microbes, such as *Acetobacter*, *Clostridium* and *Propionibacterium* spp[65].

Volatile esters are highly important aroma compounds in fermented foods, such as wine and beer[64]. The highest number of positive correlations between esters and species are with *Brettanomyces* spp. (Fig. 10), highlighting these species as key flavour producers in WK. Patel et al., observed correlations between *Br. bruxellensis* and the ester, 2-phenylethyl acetate in WK as well[23]. The role of *Brettanomyces* spp. in fermented beverages has been discussed before, which can range from essential for good flavours (e.g.,

in lambic beers) to a source for off-flavours (e.g., in wine)[66,67]. The yeast *Pi. fermentans* showed negative correlations with volatile esters (Fig. 10) and might therefore negatively impact WK flavours. While volatile acids were abundant in the 48 h samples (Supplementary Fig 10), we did not observe statistically significant correlations between individual species and volatile acids, which could be due to a redundancy in metabolic capacities between species, which could be the reason for poor separation of the volatile profiles based on the community states as well (Supplementary Fig 9). It remains to be tested whether species such as the *Brettanomyces* spp. are responsible for the predicted VOC production.

## Conclusion

In this study, we conducted a comprehensive analysis of 69 WK samples, including the early and late stages of fermentation, as well as the grains. Utilizing shotgun metagenomic sequencing in conjunction with metadata, nuclear magnetic resonance (NMR) metabolomics, and gas chromatography-mass spectrometry (GC-MS) volatilomics, our research yielded several significant findings. These include: (1) detailed elucidation of the microbial composition of WK; (2) the detection of several potentially novel species, with two novel species isolated and verified; (3) identification of 13 core genera; (4) characterization of six potential community states after 48 h of fermentation; (5) insights into the metabolic capabilities and potential interactions within the WK microbial communities; (6) identification of key species that contribute to the formation of the volatile organic compound profile; and (7) enhanced understanding of the kefir fermentation process, such as the effect of drying WK grains on microbial composition and grain growth.

## Materials and methods
### Sample acquisition

A total of 69 water kefir (WK) grains were sourced from private and commercial WK producers from November 2020 until June 2021. Individuals were asked to provide kefir grains either dried or fresh and double bagged. Information about their first fermentation parameters, such as previous fermentation durations, aerobic or anaerobic setups, and substrates used, was collected. The first fermentation is generally carried out with the WK grains to create a base product, while a second fermentation can be added to flavour the base product.

### Water kefir media preparation

A sterile fig extract was prepared as described before with the following modifications[12]. 500 g of dried figs were cut into small pieces and soaked in 1 L of tap water for 45 min while shaking at 100 rpm. Fig-water was poured through a sieve and further particles were removed by centrifugation at 17,000 × *g* for 1 h. The fig extract was vacuum-filtered through a 0.2 μm membrane and stored at 4 °C until use. A sucrose solution was prepared by adding 88 g/L sucrose to tap water and autoclaving at 121 °C for 15 min. 15 ml of sterile fig extract were added to 135 ml of 88 g/L sucrose solution (final concentrations: 80 g/L sucrose, 10% fig extract) and 9 g (60 g/L) of kefir grains, the inoculum, were added.

### Water kefir fermentation

WK fermentations were carried out under aseptic, aerobic conditions in sterile glass bottles. Glass bottles were closed with screw caps containing 0.2 μm PTFE membranes and incubated at 21 °C for 48 h. After 48 h the ferment was poured through a sterile sieve and rinsed with autoclaved tap water. Grains were re-used to start a new fermentation, as described above. As recommended previously[12], a minimum of two pre-fermentations were carried out to allow the grains to recover from transport to the research centre, before the fermentations for sample collection were carried out.

For each of the 69 WK grains, experiments were performed in two sequential biological replicates. The first biological replicate was grown for 48 h and samples were taken at 8 h and 48 h. After 48 h, the grains were rinsed with sterile tap water and reused to start the second replicate and

samples were taken again after 8 h and 48 h. Grain samples for shotgun sequencing were only taken after the second 48 h fermentation. If there was a sufficient grain growth after the first 48 h fermentation, then excess grains were taken to test the freeze tolerance of WK grains (see Supplementary Methods for additional details).

### Sample collection

15 ml of WK or uninoculated media (functioning as media, extraction, and library preparation control) were collected for DNA extraction (two times 69 WK at 8 h and 48 h; 3 times media at 0 h, 8 h and 48 h; 285 samples total). The liquid was centrifuged at 4 500 × g for 30 min. The supernatant was discarded, the pellet flash frozen in liquid nitrogen and stored at −20 °C until DNA extraction. Samples for VOC and NMR analysis were collected in 2 ml screw cap tubes, flash frozen in liquid nitrogen, and stored at −80 °C until analysis. Kefir grains were taken after the WK was sieved and rinsed with sterile tap water. Kefir grains for all 69 WK were flash frozen in liquid nitrogen and stored at −20 °C until DNA extraction.

### DNA extractions

DNA extractions were performed with the DNeasy PowerSoil Pro Kit (Qiagen, 47016), according to manufacturer instructions. DNA was extracted from the pellet of 15 ml WK liquid (276 samples), 15 ml media (9 samples; controls) or ~100 mg kefir grains (69 samples). For the DNA extraction from kefir grains, additional 1.5 mm Zirconium beads (Merck, Z763799) were added to the PowerBead Pro Tube.

### Library preparations and sequencing

Libraries for sequencing were prepared for all of the above extracted samples with the Nextera XT DNA Library Preparation Kit (Illumina, FC-131-1096) and Nextera XT Index Kit v2 (Illumina, FC-131-200X) according to manufacturer instructions. DNA quantity was checked using the Qubit dsDNA HS Assay Kit (Invitrogen, Q33231) and DNA integrity was checked using the Agilent High Sensitivity DNA Kit (Agilent, 5067-4626). Sequencing was carried out in house on the NextSeq500 using the 300-cycle High Output v2 kit.

### Read QC and MAG assembly

MetaWRAP[68] was used for read QC, host decontamination (hg38), and MAG assembly. Bacterial high quality MAGs were obtained with >90% completeness and <5% contamination by performing a co-assembly of the five metagenomics samples of each WK. CheckM[69] steps were disabled in MetaWRAP to obtain fungal MAGs. Fungal MAGs were identified and quality checked using BUSCO (reference databases: ascomycota_odb10) and only fungal MAGs with ≥70% completeness were retained[70]. Bacterial MAGs were assigned to species using GTDB-TK (v. 1.3.0; ref. 71). If bacterial MAGs could not be assigned to a species using GTDB-TK, then FastANI[72] was used with a custom database containing NCBI GenBank and RefSeq genomes for the relevant genus. Fungal MAGs were assigned to species using the FastANI approach from above with a custom database.

### Taxonomic profiling

Taxonomic profiling was performed using inStrain[73] with a custom WK database. To create the custom WK data base, Kaiju[74] was used to detect genera present in the WK metagenomes. For genera that made up >1% of the reads in at least one of the samples or were detected with at least one MAG, all corresponding reference genomes from NCBI were downloaded. Bacterial reference genomes and all bacterial MAGs from this study (Supplementary Data 1, NCBI BioProject: PRJNA977472) were combined and dereplicated using dRep[75] with an ANI value of 95%. Fungal reference genomes were manually curated using FastANI[72] to identify genomes with >98% shared ANI (e.g., subspecies). The dereplicated inStrain custom database contained 1826 species representative genomes. NCBI accession numbers and MAG names of the species representative genomes are given

in Supplementary Data 1. To count a species as present, at least 35% breadth was required, as well a minimum ratio of 0.75 for the expected breadth to observed breadth. Relative species abundance was calculated based on genome coverage based abundance. For additional information on method optimisation see Supplementary Results and Discussions.

## Metabolite analysis

For metabolite analysis and volatile organic compound analysis 15 water kefirs were chosen with one sample for 8 h and 48 h each, as well as 3 media controls (total of 33 samples).

Samples were defrosted on a roller for 40 min at room temperature, and were then centrifuged at 7870 g for 5 min at 4 °C. The supernatant was collected and filtered through washed 3 kDa Ultra centrifugal filters for 35 minutes at 14480 g (Sigma-Aldrich, Merck KGaA, Darmstadt, Germany). The filtrates were then frozen at −20 °C until further analysis.

On the day of analysis, samples were defrosted and 540 μL was mixed with 10 μL sodium trimethylsilyl [2,2,3,3-2H4] propionate (TSP) (0.05 g/4 mL D$_2$O) and 60 μL deuterium oxide (D$_2$O). Spectra were acquired from a 600-MHz Varian NMR Spectrometer (Varian Limited, Oxford, United Kingdom), using the first increment of a nuclear Overhauser enhancement spectroscopy pulse sequence at 25 °C. Spectra were acquired at 16,384 complex data points and 128 scans. Water suppression was achieved during the relaxation delay (2.5 s) and the mixing time (100 ms). All 1H-NMR sample spectra were referenced to TSP at 0.0 parts per million (ppm) and processed manually with the Chenomx NMR Suite (version 7.7) by using a line broadening of 0.2 Hz, followed by phase and baseline correction. A total of 29 metabolites were identified and quantified based on the Chenomx 600-MHz Library and The Human Metabolome Database (HMBD).

## Volatile organic compound analysis

2 g of WK liquid, 0.6 g of sodium chloride and 20 μl of internal standard (4-methyl-2-pentanol at 50ppm) was added to a 20 ml screw capped SPME vial. The SPME fibre was exposed to the headspace above the samples for 30 min at depth of 1 cm at 40 °C. Sample introduction was accomplished using a Gerstel MPS Autosampler.

A single 50/30 μm Carboxen™/divinylbenzene/polydimethylsiloxane (DVB/CAR/PDMS) fibre was used. The SPME fibre was exposed to the headspace above the samples for 30 min at 40 °C, then retracted and injected into the GC inlet and desorbed for 3 min at 250 °C. Injections were made on a Shimadzu 2010 Plus GC with an Agilent DB-624 UI (60 m × 0.32 mm × 1.8 μm) column using a split/splitless injector in a splitless mode. A merlin microseal was used as the septum. The temperature of the column oven was set at 40 °C, held for 5 min, increased at 5 °C/min to 230 °C then increased at 15 °C/min to 260 °C, held for 5 min yielding at total GC run time of 65 min. The carrier gas was helium held at a constant flow of 1.2 ml/min. The detector was a Shimadzu TQ8030 mass spectrometer detector, ran in single quad mode. The ion source temperature was 220°C and the interface temperature was set at 260 °C. The MS mode was electronic ionization (70 v) with the mass range scanned between 35 and 250 amu. Compounds were identified using mass spectra comparisons to the NIST 2014 mass spectral library, a commercial flavour and fragrance library (FFNSC 2, Shimadzu Corporation, Japan) and an in-house library created using authentic compounds with target and qualifier ions and linear retention indices for each compound using Kovats index[76]. Retention indices were matched against peer reviewed publications and authentic standards where possible to confirm compound identification. Spectral deconvolution was also performed to confirm identification of compounds using AMDIS. Batch processing of samples was carried out using MetaMS[77]. MetaMS is an open-source pipeline for GC-MS-based untargeted metabolomics. An autotune of the GCMS was carried out prior to the analysis to ensure optimal GCMS performance. A set of external standards (1-butanol, dimethyl disulfide, butyl acetate and cyclohexanone at 10ppm) was run at the start and end of the sample set and abundances were compared to known amounts to ensure that both the SPME extraction and MS detection was performing within specification.

## Statistics and reproducibility

All data processing, statistical analysis and plotting was done in R. Relative abundance was calculated as taxonomic abundance[78], based on the relative coverage of a species within the sample. Plots were created using *ggplot2* and *pheatmap*. Hierarchical clustering was done with default clustering parameters ("complete"), with the exception of correlation analysis plots, in which "ward.D" was used. Boxplots show the first and third quartiles, median, and whiskers indicate 1.5 times the interquartile range (IQR). Alpha diversity measures were calculated as in ref. [78]. Beta diversity calculations and PCoA plots were created using the *Vegan* package. *P*-values in Spearman correlations were Benjamini-Hochberg adjusted. *P*-values of <0.05, <0.01, and <0.001 are flagged with one, two, and three stars (*, **, and ***), respectively. Species co-occurrence was calculated in R using the package *cooccur*[79]. Multiple Factor Analysis (MFA) was calculated using the package *FactorMineR*[80].

## Reporting summary

Further information on research design is available in the Nature Portfolio Reporting Summary linked to this article.

## Data availability

WK shotgun metagenomics sequences, fungal and bacterial MAGs have been made available under the BioProject number PRJNA977472. All other data used for the generation of the figures and results in this study is available in Supplementary Data 1 in the online version of this manuscript.

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

## Acknowledgements

The authors would like to thank the staff maintaining the Teagasc high-performance computing cluster. The P.D.C. laboratory receives its funding from the European Union's Horizon 2020 Research and Innovation Programme through the MASTER project with grant number 818368. It is also supported by Science Foundation Ireland (SFI) under grant number SFI/12/RC/2273_P2 for APC Microbiome Ireland, and in collaboration with the Irish Department of Agriculture, Food and the Marine through grant number SFI/16/RC/3835 for the VistaMilk project. Furthermore, the laboratory benefits from funding by Enterprise Ireland and various industry partners for the Food for Health Ireland (FHI)-3 project under grant number TC/2018/0025, as well as from the Institute for the Advancement of Food and Nutritional Sciences with grant number NA-AGFOODDEVELAUTH-20201216. L.B. and X.Y. would also like to acknowledge funding from HRB/SFI (USIRL-2019-1).

## Author contributions

S.B. and P.D.C. conceived and designed the study. S.B., I. S., and X. Y. performed the experiments and analysed the data. S.B. and P.D.C. wrote the manuscript. S.B., I. S., X. Y., L.B., K.K., and P.D.C. reviewed and edited the manuscript. P.D.C. supervised the study. L.B., K.K., and P.D.C. secured the necessary funding.

## Competing interests

The authors declare the following competing interests: The laboratory led by P.D.C. has received funding from Friesland Campina, PrecisionBiotics

Group, PepsiCo, and Danone. P.D.C. has been supported by PepsiCo, Yakult, and H&H to attend and present at various scientific meetings and conferences. P.D.C. also holds the position of Chief Technical Officer and is a co-founder of SeqBiome.
