## [Transparent Peer Review file · Communications Biology]

The core microbiomes and associated metabolic potential of water kefir as revealed by pan multi-omics

Corresponding Author: Professor Paul Cotter

This manuscript has been previously reviewed at another journal. This document only contains information relating to versions considered at Communications Biology.

Version 0:

Reviewer comments:

Reviewer #1

(Remarks to the Author)

This paper presents a large-scale study of water kefir, a fermented product made from sugar water and kefir grains as inoculum. The study is based on the collection of 69 kefir grains from some twenty countries, which were analyzed by shotgun metagenomics. The authors used a sub-sample of these kefirs for a metabolic study including metabolomics and volatilomics.

The section on the analysis of the microbial composition of kefirs is very well handled and presents data carried out in a comparable way, which was not possible by compiling the data carried out by different methods in the publications. It therefore provides an excellent basis for analyzing the complex microbial composition of WK.

Building up a collection of 69 kefirs is a real strength in this work, and it's also a real challenge to put them back to work in order to study their metabolism. In this study, it appears that a number of kefirs no longer replicate, particularly those sent in dried form. Kefir is a fragile ecosystem, and people often report their loss or illness on social networks. As a result, kefirs that are no longer developing generally evolve towards decline and loss, although sometimes they can recover their metabolic activities after a number of transfers without anyone knowing exactly what happened.

Thus, in this study, it appears that dried kefirs, a significant number of which no longer replicate, also have a lower biodiversity than others. This potential loss in biodiversity could explain their lack of growth. Furthermore, even if they still show residual metabolic activity, it is likely that this activity is different from that of a healthy kefir. It is also likely that these kefirs needed more time to adapt to the basic growth conditions used in this paper (fig + sucrose), compared with a history where other elements were added to the medium. It would have been necessary to revive them with a larger number of replications. Indeed, Authors indicate line 356 "The decrease of the pH during the fermentation was used as an indicator of recovery of the grains after shipping and of a successful fermentation". This only means that some LAB were present and have retained acidifying activity but not that the WK has been revitalized. Issue related to this fact are developed below.

It is therefore advisable to process grains (dried and fresh) with/without growth separately in the study, particularly in terms of metabolic activity, so as not to bias the results of kefirs in good physiological state.

Reference:

some of the data in this article were published in Cell by Carlino et al. 2024, in particular the composition of Wks. It seems, however, that they have been reanalyzed more precisely. Perhaps this should be mentioned and the reference added ?

Specific comments

Sampling.

A remarkable feature of this study is the number of kefir samples collected. 69 WK were collected from some twenty countries. Descriptors are given from self-reported data. As a result, there is some uncertainty as to how to interpret these descriptors, such as aerobic versus anaerobic fermentation types. It would be useful to clarify what this means, especially for

scientists who are not accustomed to producing water kefir at home. And finally, how important is this parameter?

According to the practices reported, a wide variety of production histories emerge, as shown in Figure 1, both in terms of the type of sugar added, the addition of certain ingredients and notions of fermentation practices (aerobic anaerobic, usual fermentation time between 24 and over 72 hours....).

- In addition to the type of sugar added, the quantity of sugar that was added by the donors appears as an important data. This is a highly variable factor in kefir production, particularly among private individuals, where it generally ranges from 40 to 60 g, but can sometimes drop to 30 g or even 20 g when less alcoholic kefir is desired.

- There are also practices where individuals replicate kefir in a different condition from that in which they produce the beverage.

Have you included these two questions (amount of sugar, direct or 2-step fermentation) in your kefir donor questionnaire so that they can be added here?

General fermentation characteristics

The grains were restarted by at least 2 transfers (but sometimes (?) only 2, as for WK008 and WK069, which still did not replicate afterwards, see below). Subsequently, according to Fig 1g, a significant number of grains remain without growth, or even with a loss of mass, indicating that they have not recovered their normal activity. It appears that around half of the dried grains have not regained significant growth (less than 10% grain growth), compared with 15% of the fresh grains. The statement of an average increase of 50% (line 131) does not seem to reflect the reality of active kefir. Indeed, if the samples with less than 10% growth are removed, 100% weight gain for fresh grains is obtained (but only 50% for dried grains showing a less clear-cut threshold between growing and not-growing grains).

Line 100: the number of kefir samples taken at 72h and over should be increased by 5 or 6 according to fig 1c.

Fig 1f : it is not clear what the second and fourth boxes represent

Metabolite analysis, Flavour analysis

“throughout” reported at least line 268, 270, fig 6, 302 and 426 (discussion), supplementary Fig S6 S7 S8...: the authors cannot claim to have studied the metabolic activity of kefir throughout the fermentation by taking only two points (8 and 48h), knowing moreover that sugar fermentation is not complete (see below), and that nearly half of kefir producers stop fermentation at 72h or more.

Sugar consumption and production of alcohol and organic acids

According to figure 6 and S6, after 48 hours, less than half the saccharose has been consumed (60mM/180mM), indicating that fermentation is very incomplete in the majority of kefir. Moreover, the very high ethanol production (160mM) raises questions in view of the low sugar consumption. It appears also a very low production of lactate and the almost absence of acetate. These quantities do not seem to correspond to those usually found in WK. By the way, most commercial Wks have a very low alcohol content, in line with that of a healthy drink. These quantitative aspects were not discussed in the light of previous studies.

Concerning pyruvatoxime production, it seems very strange and the authors cannot leave this data unverified given the quantity produced here.

In conclusion, the authors should reconsider the exploitation of their results, considering the following factors:

1. A number of WK do not have optimal metabolic activity (see above).
2. Kefirs generally ferment for a minimum of 48 hours, and often for 72 hours or more. Generally speaking, fermentation is usually considered complete when all rapidly fermentable sugars have been consumed. Here, the fermentation was stopped prematurely for many (most?) of them at 48 hours.
3. The quantity of saccharose used in this work (80 g per liter) is very high, higher than those usually used in kefir production (boosting ethanol production?).

Figure S6. This figure is difficult to read because the colors are so close together.

Line 432; fig 5a is not appropriate since it indicates “Correlation analysis indicates key species influencing alpha diversity measures and fermentation characteristics” but not “complex interactions between fermentation practices and microbial compositions”.

supplementary data

Section on contaminations and unwanted species.

We agree that contaminating species can occur in kefir, but the considerations given here for both *Clostridium* and *Paenibacillus* are based on the observation of single kefir (wk0069 and wk0008), respectively and are therefore not convincing.

- WK069 was sent dried, so the long-distance argument for its delivery seems irrelevant. Moreover, in this day and age, postal deliveries are made by air, and sending a grain from Hong Kong by post doesn't necessarily take longer than from other parts of the world, or even a day or two longer.

- WK069 and wk008 are among the WK that have not returned to normal activity and are not replicating, or even losing weight specifically here. It is specified here that only 2 pre-fermentations were carried out, confirming that their study was not carried out appropriately.

Reviewer #2

(Remarks to the Author)

This manuscript presents a comprehensive multi-omics analysis of water kefir, using 69 samples collected from 21 countries. The study integrates data on microbial communities, metabolomic and volatile compound profiles and fermentation characteristics. This is the first large-scale study on water kefir and adds value to the understanding of this fermented beverage.

Overall feedback: The study is rich in data and analysis and is definitely a valuable contribution to the field of science, microbiology and fermentation. However, the manuscript would benefit from a more organized presentation of the results, as well as a clearer structure to guide the reader through its findings. Several figures contain too much information, making it difficult for the reader to follow, and there are repeated elements throughout the text that could be simplified. A restructuring of the figures and discussion sections would help to present the results more clearly and in a way that emphasizes the importance of the findings.

Suggestions for figure reorganization:

Fig. 1: Overview of metadata and characteristics of the 69 water kefir samples analyzed (current format works, but consider moving the titles of the graphs to the top for clarity). It might also be useful to separate the countries by continent to provide a clearer global overview.

Fig. 2: Microbial composition at the genus and species level (perhaps move the species-level heatmap to the supplementary files, as the occupancy % graph is already sufficient for main text).

Fig. 3: Alpha and beta diversity analyses. Rather than adding letters to multiple graphs, consider combining Shannon and Simpson indices into one figure, with titles above each graph to simplify visualization

Fig. 4: Correlations between microbial species and fermentation characteristics.

Fig. 5: Metabolite profiling via NMR. Consider reducing the number of metabolites presented, focusing only on those most relevant to your discussion, as the full set of boxplots makes visualization difficult.

Fig. 6: VOC profile (currently Fig. 7). Including the class of each compound could enhance the clarity. The PCoA figure (Fig. 8) could be omitted as it reiterates findings already presented in the heatmap.

Fig.7: Correlations between species and metabolites/VOCs.

Additionally, the figure legends should be more concise; details currently included in the legends can be moved to the methods or results sections.

Discussion structure: the current discussion section repeats too much of the results and methods. I suggest reorganizing it into four main sections to better contextualize the findings:

- 1. Water kefir production practices:** discuss relevant practices, comparing dry/frozen vs. fresh samples, nutrient sources (e.g., different fruits), and anaerobic/aerobic conditions. It may be helpful to bring in some regulations about water kefir being marketed as a non-alcoholic beverage (<1.2% alcohol in Europe, <0.5% in the US).
- 2. Water kefir as a source of novel species:** highlight the discovery of novel species and how these findings could benefit the food industry (e.g., the Bifidobacterium you isolated). Draw parallels to other emerging novel fermentations.
- 3. Core microbiome of water kefir:** provide a broader overview of the core microbiome, focusing on yeasts, LAB, and AAB. Consider comparing the core microbiome of water kefir with other fermented beverages like kombucha or milk kefir.
- 4. Metabolites and VOCs in water kefir:** discuss the main metabolites and VOCs, their impact on the beverage, and which species are likely responsible for producing them. This section could emphasize industrial scalability, including the importance of microbial dosage for beneficial metabolite production.

Section-specific comments

Introduction:

-Please revise the entire section for clarity. For example, lines 80-81 mention "higher residual sugar concentration" - compared to what? Also, clarify whether the four strain communities were isolated from water kefir samples (if information is available). Including the year of the patent would also be helpful.

Results:

-The figure structure can be revised as outlined earlier.

-Standardize the meaning of "NA," which is used inconsistently (no answer, no information, or not available?).

-Remove the sentence in lines 129-130.

-The BUSCO completeness threshold for fungal MAGs is too low. I recommend using a threshold of <70% completeness (include this in the methods).

-The section between lines 218-223 is confusing and needs revision.

-In line 238, you mention "several significant effects." Please provide specific examples.

-For current Fig. 5B and 5C, there are no stars indicating statistical significance.

-The unique sample (WK042-2-L-48) clustered in Fig. 7 could be highlighted in the text as it appears different from the others (rich in *Z. mobilis*, high ester concentration, etc?).

Discussion:

- Consider revising the discussion as previous suggested structure to provide more coherence and avoid repetition.

Materials and Methods:

- Include the year or period during which samples were collected.
- Mention the number of samples analyzed in each section (e.g., DNA extraction - grain & liquid, metabolite analysis, VOC analysis...).
- Address the BUSCO completeness threshold for fungal MAGs.
- Clarify whether the fungal MAGs will be accessible.

Supplementary material:

- Table 1 should be moved to the supplementary material.
- The database created in inStrain seems excellent - congratulations. - I was surprised that freezing samples did not impact diversity. How many samples were evaluated? Is there a reference to support this claim? To suggest that freezing is preferable to freeze-drying, it would be necessary to analyze the same grains under both conditions.
- The co-occurrence section needs clarification - who is co-occurring with whom?
- The alpha diversity graphs (e.g., Fig. S8) could benefit from added titles for better readability. You may also consider keeping only statistically significant factors, not all of them.
- The heatmap in Fig. S11 is difficult to interpret. Is it necessary to include this figure in its current form?

The manuscript presents insights into the microbial ecology and metabolic potential of water kefir. With improved structure and presentation, the paper has the potential to impact research on fermented beverages significantly. I congratulate the authors on the large dataset they have compiled.

Version 1:

Reviewer comments:

Reviewer #1

(Remarks to the Author)

As written in the first report of this paper, "this paper presents a large-scale study of water kefir, a fermented product made from sugar water and kefir grains as inoculum. The study is based on the collection of 69 kefir grains from some twenty countries, which were analyzed by shotgun metagenomics. The authors used a sub-sample of these kefir grains for a metabolic study including metabolomics and volatilomics. The section on the analysis of the microbial composition of kefir is very well handled and presents data carried out in a comparable way, which was not possible by compiling the data carried out by different methods in the publications. It therefore provides an excellent basis for analyzing the complex microbial composition of WK."

However, the second part of the work, concerning the metabolism of water kefir, has not been properly processed, which limits its relevance. Indeed, the study of such a large number of different water kefir samples can only be done in a relatively superficial manner or would then have required a truly enormous amount of work. It was chosen here to study sugar consumption and metabolite production with two biological replicates (a real minimum) after 8h and 48 hours, i.e. an early and medium stage of fermentation (and not late fermentation as indicated in the conclusion). Moreover, the kefir used were not all in the same physiological state, since some were developing normally, while others no longer show any growth. They did indeed have acidifying metabolic activity, and produce a certain number of metabolites, but this type of WK could not be considered as representative of healthy WK and will never be able to be used in subsequent production, as it is impossible to replicate.

Beyond these restrictions, the results obtained undoubtedly show a great diversity in metabolite production from a wide range of kefir grains. This diversity and the probable metabolic redundancy in the different kefir samples did not allow significant conclusions to be drawn (such as the production of metabolites of interest as function of particular species). The two examples given here, a correlation between branched amino acid production (technological interest?) in *Acetobacter orientalis* and volatile acid production in *Brettanomyces* spp. (infrequent), are of relatively limited interest.

Finally, the production of pyruvatoxime is very intriguing, and the discussion of its production pathway is unconvincing. In terms of assay reliability, although NMR measurements are considered reliable, the interpretation of its spectra is not straightforward and may be subject to artifacts, i.e. some products produce several peaks, peaks belonging to different compounds may overlap... As far as the production route is concerned, the explanation given is not convincing. While ammonia monooxygenase is potentially present in *Komagataeibacter*, this relatively rare species is not present in most samples containing pyruvate oxime.

Heterotrophic nitrification is a process described in the environment, and it is not certain that the conditions are right for it to occur in food products (example in the literature?), and in particular in the case of water kefir, where for example there is no ammonia available as a base substrate, especially at the levels of pyruvatoxime potentially detected here. Verification by an alternative method seems necessary.

If this compound is really present, it is necessary to provide references concerning its presence in food (or its detection for the first time?) and the potential associated risks, knowing that it would be present on average at 18 mM and up to 48 mM in samples at 48h.

In conclusion, the article is of real interest in terms of studying the biodiversity of water kefir, and deserves to be published despite the aspects I have highlighted in terms of metabolic analysis. However, it is necessary to clarify the point concerning pyruvatoxime, whose unexpected presence at a high level in WK is insufficiently established and discussed.

Reviewer #2

(Remarks to the Author)

The authors have addressed the suggestions provided, and the manuscript shows improvement. However, there are still several repetitions throughout the text that could be replaced to enhance readability. A thorough revision of the entire manuscript is recommended (some examples provided below)

Specific comments:

Introduction

-Line 26: change 'were' to 'was'

-Line 34: 'show'

-Line 42: why did you add 'the inoculum' in the text?, if subsequently you say that serve as inoculum...

-Here is one example of repetition in Line 55-59 that could be rephrase (suggestion): 'To date, several studies have investigated either individual WKS [15, 22-26] or a small number of them [12, 14, 27-29] to explore their microbial composition. These studies have identified LAB, acetic acid bacteria (AAB), and yeasts as the most common taxa in WK, with Bifidobacterium spp. and Zymomonas mobilis being among the most abundant microorganisms detected [12, 14, 15, 22-29]'

Results

-Line 147-148: this sentence is missing a verb: "were successfully"?

-Line 290-292: reduce repetitions of 'VOC' and 'WK'.

-Figures: While the journal guidelines do not specify a limit on the number of figures, 10 figures seems a lot.

Discussion

Water kefir production practices

-Line 332-333: avoid repeating 'diverse set of' in the same sentence.

-Line 340-341 & 347 & 348: repetitive sentences 'there was a relatively even split between aerobic and anaerobic WK fermentations among the participants'

-Line 352-353: And what are the findings of this study by Laureys et al?

-Line 357-361: Revise to reduce repetition of 'inocula' and 'grain growth'

Source of novel species

-Please revise 'novel species' repetition in the first sentence.

-Nice work publishing the two new Bifidobacterium species. Congrats!

Core microbiome and community states of water kefir

-Line 383: delete the parentheses

-Line 390-391: avoid repeating 'defining core microbiomes' in the same sentence

-Line 398: Bifidobacterium and Zymomonas were left out of the parentheses

-Line 398-401: revise this sentence

-Line 402: what kind of other fermented foods? Could you mention here?

-Line 403-407: long sentence, please review. Maybe divide into two sentences.

Metabolites and volatile organic compounds in water kefir

-Line 446-448: you can add as legislation/references instead of mentioned the link. And missed 'respectively' in your sentence.

-Line 488-491: several 'Brettanomyces spp.'

-Line 494: 48h, instead of 'hr'

-Line 498-499: review this sentence

Referee expertise:

Referee #1: Microbiology of Kefir fermentation

Referee #2: Omics of microbial fermentation

Reviewers' comments:

#	Reviewer #1 (Remarks to the Author):	Responses by Authors
1	This paper presents a large-scale study of water kefir, a fermented product made from sugar water and kefir grains as inoculum. The study is based on the collection of 69 kefir grains from some twenty countries, which were analyzed by shotgun metagenomics. The authors used a sub-sample of these kefirs for a metabolic study including metabolomics and volatilomics. The section on the analysis of the microbial composition of kefirs is very well handled and presents data carried out in a comparable way, which was not possible by compiling the data carried out by different methods in the publications. It therefore provides an excellent basis for analyzing the complex microbial composition of WK.	Thank you for your careful and thoughtful review of the manuscript. Your feedback is much appreciated. We hope to have addressed your suggestions to your satisfaction and improved the manuscript accordingly. The below mentioned line numbers refer to the lines number in "Simple Markup" mode in the revised manuscript.
2	Building up a collection of 69 kefirs is a real strength in this work, and it's also a real challenge to put them back to work in order to study their metabolism. In this study, it appears that a number of kefirs no longer replicate, particularly those sent in dried form. Kefir is a fragile ecosystem, and people often report their loss or illness on social networks. As a result, kefirs that are no longer developing generally evolve towards decline and loss, although sometimes they can recover their metabolic activities after a number of transfers without anyone knowing exactly what happened. Thus, in this study, it appears that dried kefirs, a significant number of which no longer replicate, also have a lower biodiversity than others. This potential loss in biodiversity could explain their lack of growth. Furthermore, even if they still show residual metabolic activity, it is likely that this activity is different from that of a healthy kefir. It is also likely that these kefirs needed more time to adapt to the basic	The aim was to study water kefir as broadly as possible. We believe that lowering of the pH is the most practical marker for our study to show re-established microbial activity after transport and to use as an inclusion criterion in the study. The lowering of the pH in water kefir is important, as a pH less than 4.4 is required by e.g. Food Safety Authority of Ireland to "prevent the growth of Listeria monocytogenes and toxin formation by Clostridium botulinum" in water kefir (Guidance Note 37, Food Safety Authority of Ireland, 2021). The use of grain growth as a marker of recovery would not have been practical of this study, as grains often did not regain grain growth even after repeated pre-fermentations, despite showing clear signs of fermentation activity. For example, after this study we have cultured WK069 repeatedly for nearly 3 months (unpublished data), but could not regain any grain growth.

	growth conditions used in this paper (fig + sucrose), compared with a history where other elements were added to the medium. It would have been necessary to revive them with a larger number of replications. Indeed, Authors indicate line 356 “The decrease of the pH during the fermentation was used as an indicator of recovery of the grains after shipping and of a successful fermentation”. This only means that some LAB were present and have retained acidifying activity but not that the WK has been revitalized. Issue related to this fact are developed below.	
3	It is therefore advisable to process grains (dried and fresh) with/without growth separately in the study, particularly in terms of metabolic activity, so as not to bias the results of kefir in good physiological state.	Thank you for the suggestion. We have added shipping information to the figures “Figure 9 Heat map detected volatile organic compounds” and “Figure S 6 Heat map detected metabolites” and replaced the figures in the manuscript. While we can see changes in the microbial composition based on drying (Figure S13 and S14), we do not see any major clustering of the samples by their shipping (dried or fresh) in their metabolic and volatilomic profiles. Instead, we are seeing clustering primarily by the fermentation duration, further suggesting that the water kefir communities were active and showing comparable metabolic activity. We therefore do not think that any further processing of the data separated by dried and fresh would be beneficial for this manuscript.
	Reference:	
4	some of the data in this article were published in Cell by Carlino et al. 2024, in particular the composition of Wks. It seems, however, that they have been reanalyzed more precisely. Perhaps this should be mentioned and the reference added ?	We have added the reference and explanations in lines 130 to 133.
	Specific comments: Sampling.	
5	A remarkable feature of this study is the number of kefir samples collected. 69 WK were collected from some twenty countries. Descriptors are given from self-reported data. As a result, there is some uncertainty as to how to interpret these descriptors, such as aerobic	On the questionnaire, participants were asked if they used “tight lid (limited gas exchange)” or “filter or cloth (allowing gas exchange)” during the first fermentation, which were considered as anaerobic and aerobic fermentations, respectively. This

	versus anaerobic fermentation types. It would be useful to clarify what this means, especially for scientists who are not accustomed to producing water kefir at home. And finally, how important is this parameter?	information was added in lines 92 and 93. The importance of aerobic vs anaerobic fermentations has been discussed in the lines 341 to 350.
6	According to the practices reported, a wide variety of production histories emerge, as shown in Figure 1, both in terms of the type of sugar added, the addition of certain ingredients and notions of fermentation practices (aerobic anaerobic, usual fermentation time between 24 and over 72 hours....).  - In addition to the type of sugar added, the quantity of sugar that was added by the donors appears as an important data. This is a highly variable factor in kefir production, particularly among private individuals, where it generally ranges from 40 to 60 g, but can sometimes drop to 30 g or even 20 g when less alcoholic kefirs are desired. - There are also practices where individuals replicate kefirs in a different condition from that in which they produce the beverage. Have you included these two questions (amount of sugar, direct or 2-step fermentation) in your kefir donor questionnaire so that they can be added here? 	Participants were only asked about their “first fermentation” practices and not about the second fermentation, which is generally used to flavour the base product from the first fermentation. We have added that the questionnaire asked about the first fermentation in lines 517 to 520. The questionnaire asked about “kind of sugar & concentration” used. Unfortunately, the sugar concentration was too often missing or given in an unhelpful way (e.g. amount of sugar without fermentation volume), so that we decided not to include the sugar concentration information in the study.
	General fermentation characteristics	
7	The grains were restarted by at least 2 transfers (but sometimes (?) only 2, as for WK008 and WK069, which still did not replicate afterwards, see below). Subsequently, according to Fig 1g, a significant number of grains remain without growth, or even with a loss of mass, indicating that they have not recovered their normal activity. It appears that around half of the dried grains have not regained significant growth (less than 10% grain growth), compared with 15% of the fresh grains. The statement of an average increase of 50% (line 131) does not seem to reflect the reality of active kefir. Indeed, if the samples with less than 10% growth are removed, 100% weight gain for fresh grains is obtained (but only 50% for dried grains showing a less clear-cut threshold between growing and not-growing grains).	For more clarity, we have added the mean and median grain growth for dried and fresh grains in lines 122-124.

8	Line 100: the number of kefir samples taken at 72h and over should be increased by 5 or 6 according to fig 1c.	We have added the information for the number of samples that were previously fermented for more than 72h in lines 91 to 92.
9	Fig 1f : it is not clear what the second and fourth boxes represent	Due to space reasons, we did not add the colour descriptions to the figure, but the legend. All red violins are unfermented media control samples and blue are fermented samples. We hope to have clarified this now in line 108 and 109.
	Metabolite analysis, Flavour analysis	
10	“throughout” reported at least line 268, 270, fig 6, 302 and 426 (discussion), supplementary Fig S6 S7 S8...: the authors cannot claim to have studied the metabolic activity of kefir throughout the fermentation by taking only two points (8 and 48h), knowing moreover that sugar fermentation is not complete (see below), and that nearly half of kefir producers stop fermentation at 72h or more.	We have replaced “throughout” with “during the sampled time points of” in the manuscript and the supplements.
	Sugar consumption and production of alcohol and organic acids	
11	According to figure 6 and S6, after 48 hours, less than half the saccharose has been consumed (60mM/180mM), indicating that fermentation is very incomplete in the majority of kefir. Moreover, the very high ethanol production (160mM) raises questions in view of the low sugar consumption. It appears also a very low production of lactate and the almost absence of acetate. These quantities do not seem to correspond to those usually found in WK. By the way, most commercial Wks have a very low alcohol content, in line with that of a healthy drink. These quantitative aspects were not discussed in the light of previous studies.	Thank you for pointing this out. We have added additional descriptions of results and discussions regarding sucrose, lactate, acetate, and ethanol in the lines 258 – 264 and 433 – 452.
12	Concerning pyruvatoxime production, it seems very strange and the authors cannot leave this data unverified given the quantity produced here.	NMR analysis is very reliable and we are confident in the NMR based detection of pyruvatoxime. Therefore, we did an additional literature review and provided an additional discussion on pyruvatoxime in lines 453 to 466. Lines 274 – 276 (initial submission) were removed.

	In conclusion, the authors should reconsider the exploitation of their results, considering the following factors:	
13	1. A number of WK do not have optimal metabolic activity (see above).	Please see responses to comment 2 and 3.
14	2. Kefirs generally ferment for a minimum of 48 hours, and often for 72 hours or more. Generally speaking, fermentation is usually considered complete when all rapidly fermentable sugars have been consumed. Here, the fermentation was stopped prematurely for many (most?) of them at 48 hours.	We have added further result descriptions and discussions regarding the remaining sucrose in lines 258 – 262 and 433 – 438.
15	3. The quantity of saccharose used in this work (80 g per liter) is very high, higher than those usually used in kefir production (boosting ethanol production?).	Our chosen sucrose concentration (80 g/L) is in line with what is commonly used in the literature (e.g. 50g/L [1], 62.5g/L [2, 3], 80g/L [4] , 100 g/L [5], 250g/L [6]) and reported for private use in [7] (110, 90, and 60 g/L). As mentioned above, we have added additional discussions on sucrose and ethanol in lines 433 – 452.
16	Figure S6. This figure is difficult to read because the colors are so close together.	We have updated Figure S6 (now Figure S8) by splitting the figure into four subfigures for the classes organic acids, alcohols, sugars, and others. The legend has been updated accordingly.
17	Line 432; fig 5a is not appropriate since it indicates “Correlation analysis indicates key species influencing alpha diversity measures and fermentation characteristics” but not “complex interactions between fermentation practices and microbial compositions”.	Yes, we believe that there are complex interactions between fermentation practices, individual species, and overall microbial composition. We have changed the legend for the figure (now Figure 6) to “Correlation analysis indicates species correlating with alpha diversity measures and fermentation characteristics”. Lines 248-249. We have also updated the discussion to ...“we observed influences of fermentation practices (self-reported data from grain providers) on alpha diversity measures that persisted throughout the shipment of the grains and the pre-fermentations (Figure S 5), highlighting complex interactions between fermentation practices, individual species, and microbial compositions (Figure 6 and Figure S 5). Lines 427 – 431.
	Supplementary data	

18	Section on contaminations and unwanted species. We agree that contaminating species can occur in kefir, but the considerations given here for both Clostridium and Paenibacillus are based on the observation of single kefirs (wk0069 and wk0008), respectively and are therefore not convincing.	We have not seen high enough numbers of Clostridium or Paenibacillus to investigate their occurrence in more detail. We therefore see these particular occurrences as individual case studies of a grain that was particularly long in the mail (WK069) and a grain that was dried particularly long (WK008), which might warrant further investigations in the future. We have renamed this section to highlight that these are individual observations. New section title: "Observation of individual cases of contaminations and undesired species"
19	- WK069 was sent dried, so the long-distance argument for its delivery seems irrelevant. Moreover, in this day and age, postal deliveries are made by air, and sending a grain from Hong Kong by post doesn't necessarily take longer than from other parts of the world, or even a day or two longer.	For some unknown reason WK0069 was in the mail for 83 days (08.04.2021 to 30.06.2021; date stamp by the post and delivery day in the lab) and therefore stood out as being particularly long in the mail. To provide more context, we have added the additional information in Sup. line 219.
20	- WK069 and wk008 are among the WK that have not returned to normal activity and are not replicating, or even losing weight specifically here. It is specified here that only 2 pre-fermentations were carried out, confirming that their study was not carried out appropriately.	Please see above regarding pH and grain growth as markers of recovery. While two pre-fermentations were performed with WK069, several pre-fermentations were performed with WK008 (Sup. lines 229 – 230) before any lowering of the pH was observed.

#	Reviewer #2 (Remarks to the Author):	Responses by Authors
1	This manuscript presents a comprehensive multi-omics analysis of water kefir, using 69 samples collected from 21 countries. The study integrates data on microbial communities, metabolomic and volatile compound profiles and fermentation characteristics. This is the first large-scale study on water kefir and adds value to the understanding of this fermented beverage.	Thank you for your careful and thoughtful review of the manuscript. Your feedback is much appreciated. We hope to have addressed your suggestions to your satisfaction and improved the manuscript accordingly. The below mentioned line numbers refer to the lines number in “Simple Markup” mode in the revised manuscript.
2	Overall feedback: The study is rich in data and analysis and is definitely a valuable contribution to the field of science, microbiology and fermentation. However, the manuscript would benefit from a more organized presentation of the results, as well as a clearer structure to guide the reader through its findings. Several figures contain too much information, making it difficult for the reader to follow, and there are repeated elements throughout the text that could be simplified. A restructuring of the figures and discussion sections would help to present the results more clearly and in a way that emphasizes the importance of the findings.	Thank you for the suggestions. We have restructured the results and discussion as suggested. Please find detail descriptions of the changes below.
	Suggestions for figure reorganization:	
3	Fig. 1: Overview of metadata and characteristics of the 69 water kefir samples analyzed (current format works, but consider moving the titles of the graphs to the top for clarity). It might also be useful to separate the countries by continent to provide a clearer global overview. Fig. 2: Microbial composition at the genus and species level (perhaps move the species-level heatmap to the supplementary files, as the occupancy % graph is already sufficient for main text). Fig. 3: Alpha and beta diversity analyses. Rather than adding letters to multiple graphs, consider combining Shannon and Simpson indices into one figure, with titles above each graph to simplify visualization	Thank you for the suggestions. We have restructured many of the figures as suggested. In Figure 1 we have moved the title to the top and in a) we have organized the countries by continent and colour coded the bars accordingly. We have left Figure 2 (MAGs) unchanged, except for the increased fungal completes of 70%. We have split Figure 3 into three different figures as suggested. Figure 3a and b remain unchanged. Figure 3 alpha diversity plots have become a separate figure (now Figure 4). To avoid confusing statistics, we have kept the Shannon and Simpson indices as separate

	Fig. 4: Correlations between microbial species and fermentation characteristics. Fig. 5: Metabolite profiling via NMR. Consider reducing the number of metabolites presented, focusing only on those most relevant to your discussion, as the full set of boxplots makes visualization difficult. Fig. 6: VOC profile (currently Fig. 7). Including the class of each compound could enhance the clarity. The PCoA figure (Fig. 8) could be omitted as it reiterates findings already presented in the heatmap. Fig.7: Correlations between species and metabolites/VOCs.	subfigures, but have added titles for simplified visualization. The species heatmap has been moved to the supplements as suggested and has become Figure S4. Beta diversity plots remain unchanged and have become Figure 5. We have created a separate figure for correlations between microbial species and fermentation characteristics as suggested (Figure 6). We have added titles to the metabolite boxplots as suggested and limited the plots to the most important metabolites (Figure 7). The figure with all boxplots has been move to the supplements and has become Figure S7. We have added the correlations between species and metabolites as a separate figure (Figure 8) We have added class information to the VOC heatmap as suggested. Additional shipping information was added as requested by reviewer 1. The figure has become Figure 9. We have added the correlations between species and VOCs as a separate figure (Figure 10). The PCoA figure has been moved to the supplements and has become Figure S9. The figure numbers and cross-references have been updated accordingly in the manuscript and supplements.
4	Additionally, the figure legends should be more concise; details currently included in the legends can be moved to the methods or results sections.	We have updated the figure legends according to the figure changes. It is our understanding that communications biology requests details, such as sample size and statistics, to be present in the figure legends. We have removed additional details when possible.
5	Discussion structure: the current discussion section repeats too much of the results and methods. I suggest reorganizing it into four main sections to better contextualize the findings:	Thank you for the suggestions. We have added four suggested sections as discussion subtitles to provide more structure to the reader. Additionally, we have removed repeats of results when possible.

1. Water kefir production practices: discuss relevant practices, comparing dry/frozen vs. fresh samples, nutrient sources (e.g., different fruits), and anaerobic/aerobic conditions. It may be helpful to bring in some regulations about water kefir being marketed as a non-alcoholic beverage (<1.2% alcohol in Europe, <0.5% in the US). 2. Water kefir as a source of novel species: highlight the discovery of novel species and how these findings could benefit the food industry (e.g., the Bifidobacterium you isolated). Draw parallels to other emerging novel fermentations. 3. Core microbiome of water kefir: provide a broader overview of the core microbiome, focusing on yeasts, LAB, and AAB. Consider comparing the core microbiome of water kefir with other fermented beverages like kombucha or milk kefir. 4. Metabolites and VOCs in water kefir: discuss the main metabolites and VOCs, their impact on the beverage, and which species are likely responsible for producing them. This section could emphasize industrial scalability, including the importance of microbial dosage for beneficial metabolite production.	Regarding 1.: A brief discussion of relevant practices is given in lines 335 to 352. Due to the journals word limits for results and discussions, we think it is best to leave the comprehensive discussions (>600 words) on microbial associations with grain growth and the impact of freezing on grain growth in the supplements. We have added an additional reference to how nutrient sources influence the fermentations in lines 352 – 353. Aerobic vs anaerobic fermentations are discussed in lines 340 to 350. We have added a discussion on alcohol content and regulations in the metabolite section of the discussion (lines 438 – 452). Regarding 2.: The sentences about putatively novel Pichia species was removed from the discussion, as these MAGs were <70% complete. We have added an additional reference to direct the reader towards a review discussing the use of starter cultures for the production of novel fermented foods. Lines 377 - 378. Regarding 3.: We have created a joined section entitled “Core microbiome and community states of water kefir”. We have made small changes to the core microbiome description, to highlight yeasts, AAB and LAB (lines 395 - 398). We do believe that a more in-depth comparison of the water kefir core microbiome with milk kefir or kombucha would warrant its own separate review and is out of the cope and word limits of this discussion. Regarding 4.: We have added an additional discussion of the main metabolites and their possible producers (sugars: lines 433 – 438), organic acids and alcohol: lines 438– 452). Examples on how fermentation practices could have been adapted to increase the production of e.g. organic acids and reduction of sugars have been added in lines 433 – 435 and 450 - 452.
--	--

	Section-specific comments: Introduction	
6	-Please revise the entire section for clarity. For example, lines 80-81 mention “higher residual sugar concentration” - compared to what? Also, clarify whether the four strain communities were isolated from water kefir samples (if information is available). Including the year of the patent would also be helpful.	We have clarified this section by adding that these communities were “compared to fermentations with WK grains”. We have also added additional information on the patented cultures (“isolated from water kefir” and the publication year). See lines 71-73.
	Results:	
7	-The figure structure can be revised as outlined earlier.	The figure structure has been revised as described above.
8	-Standardize the meaning of “NA,” which is used inconsistently (no answer, no information, or not available?).	We have updated lines 89 and 112 to ensure that NA always is used for “not available”.
9	-Remove the sentence in lines 129-130.	Removed the sentence.
10	-The BUSCO completeness threshold for fungal MAGs is too low. I recommend using a threshold of <70% completeness (include this in the methods).	We have filtered the fungal MAGs now with $\geq 70\%$ completeness as suggested. Figure 2 and the manuscript have been updated accordingly.
11	-The section between lines 218-223 is confusing and needs revision.	We have rephrased the section to: “Different CS were supported by genus-level beta diversity analysis (Figure 5b; ANOSIM statistic R: 0.6967), while species-level beta diversity analysis shows a less clear separation between the proposed community states (ANOSIM statistic R: 0.3807; Figure 5c).” (lines 221 - 224)
12	-In line 238, you mention “several significant effects.” Please provide specific examples.	We have added two examples to the text in lines 204-207: “decreased alpha diversity was observed in samples with previous aerobic fermentations compared to anaerobic fermentations (Figure S 4a and b) and the highest alpha diversity was observed in samples that were previously fermented for ≥ 72 h (Figure S 5c and d).”
13	-For current Fig. 5B and 5C, there are no stars indicating statistical significance.	We can see the stars in our version, but the resolution could be better. We will check with the editorial team to ensure the figures will be embedded with high resolution in the final version.
14	-The unique sample (WK042-2-L-48) clustered in Fig. 7 could be highlighted in the	We have highlighted the sample in lines 294-296.

	text as it appears different from the others (rich in Z. mobilis , high ester concentration, etc?).	“After 48h of fermentation, the Zym. mobilis dominated WK (WK042) showed a distinct VOC profile, rich in esters, and clustered separately from other WK flavour profiles (Figure 9).”
	Discussion:	
15	- Consider revising the discussion as previous suggested structure to provide more coherence and avoid repetition.	The discussion has been revised as described above.
	Materials and Methods:	
16	- Include the year or period during which samples were collected.	We have added this information in line 516.
17	- Mention the number of samples analyzed in each section (e.g., DNA extraction - grain & liquid, metabolite analysis, VOC analysis...).	Added in lines: 515, 538, 546-548, 552, 557, 561, 583-584.
18	- Address the BUSCO completeness threshold for fungal MAGs.	Increased the BUSCO completeness threshold to $\geq 70\%$ and updated figures and the manuscript accordingly.
19	- Clarify whether the fungal MAGs will be accessible.	We have added the $\geq 70\%$ complete fungal MAGs to the BioProject number PRJNA977472 and updated the manuscript in line 658.
	Supplementary material:	
20	-Table 1 should be moved to the supplementary material.	Table has been moved to the supplements. Cross-references and citations have been updated.
21	-The database created in inStrain seems excellent - congratulations. - I was surprised that freezing samples did not impact diversity. How many samples were evaluated? Is there a reference to support this claim? To suggest that freezing is preferable to freeze-drying, it would be necessary to analyze the same grains under both conditions.	Thank you. We did not analyse the microbial communities after freezing or the impact of freeze-drying. We only analysed the grain growth after freezing, which was not impacted significantly. To further clarify this, we have updated the Sup. lines 138 and 147 to 148.
22	- The co-occurrence section needs clarification - who is co-occurring with whom?	We have updated Sup. lines 151, 168 – 172 and 309 to clarify that the analysis looks at the co-occurrences between pairs of species.
23	- The alpha diversity graphs (e.g., Fig. S8) could benefit from added titles for better readability. You may also consider keeping only statistically significant factors, not all of them.	As suggested, we have added titles to the Figures 7, S7, S11, S13 and S14. We prefer to keep the statistically insignificant factors, as these allow the reader a quick

		overview of abundances/concentrations for these factors.
24	- The heatmap in Fig. S11 is difficult to interpret. Is it necessary to include this figure in its current form?	Are we correct to assume that you mean Figure S5 (now Figure S6), as this is the only heatmap in the supplementary information, or are you referring to the species co-occurrence matrix in Figure S12 (now Figure S15)? We do think that both of the figures contribute to the manuscript and would prefer to keep them. We have added additional context to the figures in the legends and hope that these figures are more easily to interpret with the additional context and associated discussions.
25	The manuscript presents insights into the microbial ecology and metabolic potential of water kefir. With improved structure and presentation, the paper has the potential to impact research on fermented beverages significantly. I congratulate the authors on the large dataset they have compiled.	Thank you!

References

1. Patel, S.H., et al., *A temporal view of the water kefir microbiota and flavour attributes*. Innovative Food Science & Emerging Technologies, 2022: p. 103084.
2. Laureys, D., et al., *The Buffer Capacity and Calcium Concentration of Water Influence the Microbial Species Diversity, Grain Growth, and Metabolite Production During Water Kefir Fermentation*. Front Microbiol, 2019. **10**: p. 2876.
3. Laureys, D., et al., *Oxygen and diverse nutrients influence the water kefir fermentation process*. Food Microbiol, 2018. **73**: p. 351-361.
4. Gulitz, A., et al., *Comparative phylobiomic analysis of the bacterial community of water kefir by 16S rRNA gene amplicon sequencing and ARDRA analysis*. J Appl Microbiol, 2013. **114**(4): p. 1082-91.
5. Gulitz, A., et al., *The microbial diversity of water kefir*. Int J Food Microbiol, 2011. **151**(3): p. 284-8.
6. Laureys, D., et al., *Investigation of the instability and low water kefir grain growth during an industrial water kefir fermentation process*. Appl Microbiol Biotechnol, 2017. **101**(7): p. 2811-2819.
7. Laureys, D. and L. De Vuyst, *The water kefir grain inoculum determines the characteristics of the resulting water kefir fermentation process*. J Appl Microbiol, 2017. **122**(3): p. 719-732.

Reviewers' comments:

#	Reviewer #1 (Remarks to the Author):	Responses by Authors:
1	As written in the first report of this paper, “this paper presents a large-scale study of water kefir, a fermented product made from sugar water and kefir grains as inoculum. The study is based on the collection of 69 kefir grains from some twenty countries, which were analyzed by shotgun metagenomics. The authors used a sub-sample of these kefirs for a metabolic study including metabolomics and volatilomics. The section on the analysis of the microbial composition of kefirs is very well handled and presents data carried out in a comparable way, which was not possible by compiling the data carried out by different methods in the publications. It therefore provides an excellent basis for analyzing the complex microbial composition of WK.”	Thank you for your contribution to reviewing the revised version of the manuscript. We have made changes to the manuscript as discussed below and hope to have fully addressed all your remarks.
2	However, the second part of the work, concerning the metabolism of water kefir, has not been properly processed, which limits its relevance. Indeed, the study of such a large number of different water kefirs can only be done in a relatively superficial manner or would then have required a truly enormous amount of work. It was chosen here to study sugar consumption and metabolite production with two biological replicates (a real minimum) after 8h and 48 hours, i.e. an early and medium stage of fermentation (and not late fermentation as indicated in the conclusion).	Yes, given the larger number of WK that were obtained for this study, we had to limit the metabolic analysis to a subset of samples and time points.
3	Moreover, the kefir used were not all in the same physiological state, since some were developing normally, while others no longer show any growth. They did indeed have acidifying metabolic activity, and produce a certain number of metabolites, but this type of WK could not be considered as representative of healthy WK and will never be able to be used in subsequent production, as it is impossible to replicate.	While WK grains without growth might currently not be suited for WK production, we have included these WK in this study, as it gives us novel insights into the larger water kefir ecology. E.g. we have learned that drying of the WK grains negatively impacts future grain growth (Figure 1g) and changes the microbial composition (Figures 4c, 4d, S13 and S14), but has less of an impact on metabolic activity and the lowering of the pH (Figure 1f, 9 and S6). WK grain growth will likely be undesirable when WK is produced with starter cultures.
4	Beyond these restrictions, the results obtained undoubtedly show a great diversity in metabolite production from a wide range of kefir grains. This diversity and the probable metabolic redundancy in the different kefir samples did not allow significant conclusions to be drawn (such as the production of metabolites of interest as function of particular species). The two examples given here, a correlation between branched amino acid	We believe that the lack in significant conclusions (significant correlations between individual species and metabolites/VOCs) is nevertheless valuable knowledge for the fermented foods community, as it suggests redundancy of metabolic capacity among species. We had previously mentioned this in lines 480-483 and have additionally highlighted this in the revised version in lines 460-462.

	production (technological interest?) in Acetobacter orientalis and volatile acid production in Brettanomyces spp. (infrequent), are of relatively limited interest.	
5	Finally, the production of pyruvatoxime is very intriguing, and the discussion of its production pathway is unconvincing. In terms of assay reliability, although NMR measurements are considered reliable, the interpretation of its spectra is not straightforward and may be subject to artifacts, i.e. some products produce several peaks, peaks belonging to different compounds may overlap... As far as the production route is concerned, the explanation given is not convincing. While ammonia monooxygenase is potentially present in Komagataeibacter, this relatively rare species is not present in most samples containing pyruvate oxime.	Currently we do not have access to an alternative method for detection and quantification of pyruvatoxime. Therefore, we have moved the discussion of pyruvatoxime to the supplemental text, as suggested by the editor as well (Sup. lines 240-273). Pyruvatoxime has therefore also been removed from Figure 7 and the figure has been limited to the well-studied metabolites (acetate, ethanol, fructose, glucose, lactate, and sucrose). Pyruvatoxime is still included in the Figures S6, S7, S8, and Table S6. The reader is referred to the supplementary discussion of pyruvatoxime in lines 449-450. In response to the reviewer's comments, we now include a more critical discussion of the results and are more careful with our conclusions. For this, we have added that pyruvatoxime has not been detected in water kefir before (Sup. lines 241-243) and that a complementary method for detection is desirable (Sup. lines 243-244). Further changes are detailed below. We have added an additional tblastn search for possible AMO and POD coding genes in the WK MAGs (Table S9; Sup. lines 262-269; method in Sup. lines 38-44). The results suggest that possible AMO coding genes are present in several AAB species and Zymomonas. POD related genes were not reliably detectable (Sup. lines 262-268).
6	Heterotrophic nitrification is a process described in the environment, and it is not certain that the conditions are right for it to occur in food products (example in the literature?), and in particular in the case of water kefir, where for example there is no ammonia available as a base substrate, especially at the levels of pyruvatoxime potentially detected here. Verification by an alternative method seems necessary.	We have added that "Pyruvatoxime has been studied primarily in the context of heterotrophic nitrification, a nitrogen cycling process in the environment that is not typically associated with foods" (Sup. lines 251-252), to highlight a possible limitations of this hypothesis. In Sup. lines 271 we repeat, that a "complementary detection method is desirable".
7	If this compound is really present, it is necessary to provide references concerning its presence in food (or its detection for the first time?) and the potential associated risks, knowing that it would be present on average at 18 mM and up to 48 mM in samples at 48h.	We have conducted an additional literature search on pyruvatoxime in foods (Sup. lines 246-249). The available literature is highly limited; however, one study we identified reported the presence of pyruvatoxime in nearly all analysed malt and beer samples. Additionally, pyruvatoxime has been detected in some human samples (Sup. lines 249-250). However, there appear to be no studies specifically investigating the implications of pyruvatoxime for human health.

		Given the ongoing uncertainties surrounding pyruvatoxime in WK, we do not believe it is necessary to expand the discussion to include the broader role of oximes in human health.
8	In conclusion, the article is of real interest in terms of studying the biodiversity of water kefir, and deserves to be published despite the aspects I have highlighted in terms of metabolic analysis. However, it is necessary to clarify the point concerning pyruvatoxime, whose unexpected presence at a high level in WK is insufficiently established and discussed.	Thank you for your review of the manuscript. We hope to have addressed your comments regarding pyruvatoxime sufficiently (and have moved the associated text to the supplemental section), as discussed above.

#	Reviewer #2 (Remarks to the Author):	Responses by Authors:
1	The authors have addressed the suggestions provided, and the manuscript shows improvement. However, there are still several repetitions throughout the text that could be replaced to enhance readability. A thorough revision of the entire manuscript is recommended (some examples provided below)	Thank you for reviewing the revised version of this manuscript. We have addressed the examples you mentioned below and have checked the manuscript again to improve readability.
	Specific comments: Introduction	
2	-Line 26: change 'were' to 'was'	Changed.
3	-Line 34: 'show'	Changed.
4	-Line 42: why did you add 'the inoculum' in the text?, if subsequently you say that serve as inoculum...	We have removed "i.e., the inoculum", as it was redundant.
5	-Here is one example of repetition in Line 55-59 that could be rephrase (suggestion): 'To date, several studies have investigated either individual Wks [15, 22-26] or a small number of them [12, 14, 27-29] to explore their microbial composition. These studies have identified LAB, acetic acid bacteria (AAB), and yeasts as the most common taxa in WK, with Bifidobacterium spp. and Zymomonas mobilis being among the	Thank you for the suggestion. We have changed the text accordingly.

	most abundant microorganisms detected [12, 14, 15, 22-29]’.	
	Results	
6	-Line 147-148: this sentence is missing a verb: “were successfully”?	Thank you. “were” was added.
7	-Line 290-292: reduce repetitions of ‘VOC’ and ‘WK’.	Repetitions were reduced and the sentence was changed to “Volatile organic compound (VOC) are contributors to the flavour and their analysis, a.k.a. flavour analysis, allowed the detection of 84 compounds (Table S 8). Media samples clustered separately from fermented samples and a grouping of the WK samples by time point was discernible (Figure 9 and Figure S 9)”.
8	-Figures: While the journal guidelines do not specify a limit on the number of figures, 10 figures seems a lot.	While 10 figures seem a lot, we have actually reduced the figure content from the initial submission by removing (1) the species heatmap, (2) reducing the metabolite sub-plots from 29 to 6, and (3) removing the VOC PCoA plot. We hope that the splitting of the figures makes it easier for the reader to follow the results and discussion.
	Discussion Water kefir production practices	
9	-Line 332-333: avoid repeating ‘diverse set of’ in the same sentence.	Changed to: “WK is a fermented beverage produced with a diverse set of fermentation practices and microbes [2]”. Lines 331-332.
10	-Line 340-341 & 347 & 348: repetitive sentences ‘there was a relatively even split between aerobic and anaerobic WK fermentations among the participants’	The repeat was removed from lines 347-348.
11	-Line 352-353: And what are the findings of this study by Laureys et al?	We have added that e.g. “e.g. elevated nutrient concentrations favoured the growth of Sa. cerevisiae and Li. nagelii ” in lines 351-352.
12	-Line 357-361: Revise to reduce repetition of ‘inocula’ and ‘grain growth’	The sentences were revised to “We observed that different inocula strongly impacted grain growth. At the extremes, some grains lost mass during fermentations, while others gained more than 200% (Figure S 1; Figure 1g). Previous studies have shown that the inoculum can impact grain growth, with reported growth ranging from 4.58 to 63.82% across three different WK [14]. Associations between WK grain growth, species, and storage conditions (drying and freezing), are discussed in the Supplementary Results and Discussion” (lines 355-361).
	Source of novel species	
13	-Please revise ‘novel species’ repetition in the first sentence.	Repetitions were reduced and the sentences were rephrased to “MAGs can be used to identify putatively novel species in fermented foods [29, 42]. The commonly accepted threshold of <95% ANI for bacterial species delineation [43] allowed the detection of 18 putatively novel species (Figure 2), with their biology remaining to be explored. These previously undescribed species make up nearly 20% of the WK-associated species detected in this study, highlighting the potential for discovering novel

		organisms from this fermented beverage. We set out to validate our predictions and successfully isolated the two novel Bifidobacterium species [32], extending the number of bifidobacteria that have been isolated from WK [44, 45]" (lines 363-370).
14	-Nice work publishing the two new Bifidobacterium species. Congrats!	Thank you!
	Core microbiome and community states of water kefir	
15	-Line 383: delete the parentheses	Parentheses were removed.
16	-Line 390-391: avoid repeating 'defining core microbiomes' in the same sentence	We have rephrased the sentence to "A recent study by Custer et al., 2023 [51] has suggested that occupancy based methods are most reliable for defining core microbiomes and using genus level taxonomic information is a common approach for this in metagenomic studies [50]" (lines 387-389).
17	-Line 398: Bifidobacterium and Zymomonas were left out of the parentheses	In the parentheses the core microbiome genera of yeasts, AAB and LAB are given. Bifidobacterium and Zymomonas have therefore been left out of the parentheses.
18	-Line 398-401: revise this sentence	We have revised this sentence and the following sentence. "The here defined WK core microbiome and the average of 10.3 species per sample provides a foundation for defining what a synthetic or pitched WK community for industrial production could look like. We hope that this work will contribute towards a regulatory definition of WK, similar to the definitions that have been in place for other fermented foods, such as yoghurt and milk kefir (reviewed in [52])" (lines 396-400).
19	-Line 402: what kind of other fermented foods? Could you mention here?	We have added yoghurt and milk kefir as examples.
20	-Line 403-407: long sentence, please review. Maybe divide into two sentences.	We have split the sentence into two and rephrased: "Different definitions and approaches have been proposed to define community structures, such as community types, community states (CS), enterotypes and community signatures [53-58]. We chose to investigate CS based on a genus-level abundance-based approach after 48h of fermentation, as it closest reflects the different CS present in the consumed beverages" (lines 401-404).
	Metabolites and volatile organic compounds in water kefir	
21	-Line 446-448: you can add as legislation/references instead of mentioned the link. And missed 'respectively' in your sentence.	The links have been replaced with references. Added "respectively".
22	-Line 488-491: several 'Brettanomyces spp.'	Removed two repetitions and rephrased: "The highest number of positive correlations between esters and species are with Brettanomyces spp. (Figure 10), highlighting these species as key flavour producers in WK. Patel et al., observed correlations between Br. bruxellensis and the ester, 2-phenylethyl acetate in WK as well [23]. The role of Brettanomyces spp. in fermented beverages has

		been discussed before, which can range from essential for good flavours (e.g., in lambic beers) to a source for off-flavours (e.g., in wine) [76, 77]" (lines 473-478).
23	-Line 494: 48h, instead of 'hr'	Changed.
24	-Line 498-499: review this sentence	Rephrased: "It remains to be tested whether species such as the Brettanomyces spp. are responsible for the predicted VOC production" (lines 484-485).